# Stratum Corneum Structure and Function Studied by X-ray Diffraction

Ichiro Hatta

Division of Research, Nagoya Industrial Science Research Institute, Nagoya 464-0819, Japan; ichirohatta@gmail.com

**Abstract:** X-ray diffraction is one of the powerful tools in the study of a variety of structures in the stratum corneum at the molecular level. Resolving structural modifications during functioning is an important subject for clarifying the mechanism of operating principles in the function. Here, the X-ray diffraction experimental techniques used in the structural study on the stratum corneum are widely and deeply reviewed from a perspective fundamental to the application. Three typical topics obtained from the X-ray diffraction experiments are introduced. The first subject is concerned with the disruption and the recovery of the intercellular lipid structure in the stratum corneum. The second subject is to solve the moisturizing mechanism at the molecular level and the maintenance of normal condition with moisturizer, being studied with special attention to the structure of soft keratin in the corneocytes in the stratum corneum. The third subject is the so-called 500 Da rule in the penetration of drugs or cosmetics into skin, with attention paid to the disordered intercellular lipid structure in the stratum corneum.

**Keywords:** ceramide; cosmetics; disorder; drug; liquid; moisturizer; penetration; skin; surfactant; water

## 1. Introduction

The stratum corneum (SC) is composed of various structural elements. Each structure is related to the functionality of the SC. Under normal conditions, its main function is to provide a barrier against the penetration of harmful substances from the outside and prevent the evaporation of water to the outside. When the SC is out of normal conditions, to improve the damaged SC, it is important to clarify the relationship between structural changes and characteristics of damage. As a typical example, this paper focuses on the recovery mechanism of the damaged SC, the moisturizing mechanism with a humectant, and the penetration mechanism of small molecules in the SC studied by X-ray diffraction. This method can be extended to the other related subjects occurring in the SC.

The SC is formed by intercellular lipids composed of ceramides, free fatty acids, cholesterol, etc. and corneocytes with soft keratin fibers. In the SC the corneocytes are embedded in the intercellular lipid matrix as shown in Figure 1 as a modified bricks and mortar model [1]. To show that the water content of the SC decreases towards the surface of the skin as obtained in in vivo confocal Raman microscope (CRM) [2–4], in Figure 1. The thickness of the corneocyte is exaggerated to increase in size as it approaches the viable cells. This is consistent with results obtained by cryo-scanning electron microscopy [5], which show that, as far as the water content dependence of the average corneocyte thickness in the SC is concerned, the thickness increases linearly with increasing the water content. The SC consists typically of 12–16 layers of flattened corneocytes [6]. Water is constantly supplied from viable cells, and the same amount of water constantly evaporates from the surface through intercellular and transcellular permeation pathways.

To investigate the relationship between the structure and the function in the SC and to know the role of X-ray diffraction study in it, methods used for skin evaluation need to be compared, because these have advantages and disadvantages. Table 1 shows representative methods and their characteristics.

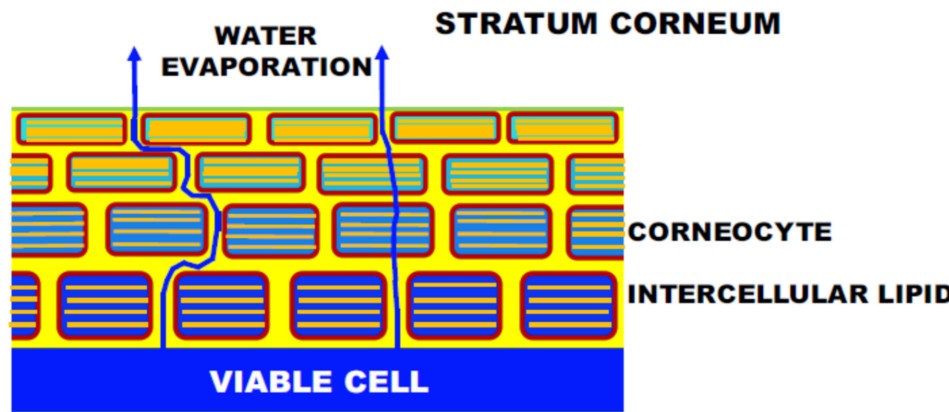

**Figure 1.** A modified brick-mortar model of the stratum corneum drawn with the contribution of water in mind. Water is constantly supplied from viable cells. Water penetrates along the tortuous pathways of the intercellular lipid matrix and across corneocytes. Finally, water constantly evaporates from the surface of the skin. As the water content decreases toward the surface of the skin, the thickness of the corneocytes is designed to decrease upward.

**Table 1.** Comparison among representative methods used for skin evaluation. They have their advantage and disadvantage, but they are complementary.

| | Measurable Sample | Distribution or Localization of Water | What Is Observed | Spatial Resolution or Structure at the Molecular Level | Reference Number When the Methods Appear in Text |
|---|---|---|---|---|---|
| X-ray Diffraction * | *ex vivo* | water storage and regulation | changes of lamellar, lateral hydrocarbon-chain packing and keratin structures | molecular arrangement | 10, 11, 12, 49, etc. |
| ATR-FTIR | *in vivo* | hard to obtain direct evidence | change of hydrocarbon-chain packing structure near skin surface | clear separation of hexagonal and orthorhombic spectra | 23, 101 |
| Confocal Raman Microscopy | *in vivo* | acquiring spectrum of water in skin depth direction | distribution of water, NMF, etc., in depth direction | ca. 3 μm | 2, 3, 4, 60, 91 |
| TEWL | *in vivo* | leakage of water through skin damage | relative variation in skin condition to external changes | ca. 1 cm intervals on skin surface | 23, 41 |
| Electrical Conductance or Capacitance Measurements | *in vivo* | water-holding capacity within skin | relative variation of water content in skin | ca. 1 cm intervals on skin surface | 41 |

* For related methods such as neutron diffraction, electron diffraction, and electron microscopy, see text.

Similar to X-ray diffraction, in the structural study of the SC, electron diffraction and neutron diffraction are used. In the diffraction studies, the intensity associated with the Fourier transform of the structures can be obtained. Therefore, if there is a periodic or regular structure in the SC, the detailed analysis of the structure is possible. The canonical resolution of the periodic structures is given by $d/n_{max}$, where $d$ is the period and $n_{max}$ is the maximum number of peak whose intensity can be observed. By the X-ray diffraction and neutron diffraction [7], the diffraction of the SC lamellar structures can be observed, e.g., the long lamellar structure whose period is about 13 nm. Likewise, by X-ray diffraction and electron diffraction, as discussed later, the hydrocarbon-chain packing structure is observable, its lattice constant is around 0.4 nm as discussed later. For these diffraction studies, an ex vivo SC sample can be used. The X-ray and neutron diffraction can be

performed close to natural condition, but the electron diffraction can be carried out in vacuum. The real image of the SC can be observed by electron microscope. Therefore, using an electron microscopy, the lamellar structure can be observed. The electron density distribution of the lamellar structure can be obtained with electron microscopes [8,9]. On a larger scale, we can observe real images of the SC with various microscopes, such as optical microscope, fluorescence microscope, etc.

The intercellular lipids in the SC form ordered structures, which are characterized in terms of two orthogonal lattice spacings: one is the lamellar-repeat spacing and another is the lattice spacing of the lateral hydrocarbon-chain packing structure. The lamellar structures are schematically shown in Figure 2: (a) long-period lamellar structure (LLS) [10–13], (b) short-period lamellar structure (SLS) [7,11,12] and (c) disordered liquid state (LIQUID). Although there is no direct evidence of the existence of the LIQUID shown in Figure 2c, the existence is inferred from the results obtained by Bouwstra et al. [11] from X-ray diffraction experiments of the human SC, i.e., in the SC after being heated to 120 °C and cooled to room temperature, clear X-ray diffraction peaks of the LLS appeared. This fact indicates that a disordered lipid mixture that can emerge the LLS exists originally. The hydrocarbon-chain packing structures are shown in Figure 3: (a) orthorhombic hydrocarbon-chain packing structure (ORTHO) [10,14–22], (b) hexagonal hydrocarbon-chain packing structure (HEX) [17–19] and (c) disordered liquid state (LIQUID) [10,16,17]. The LLS spacing is about 13 nm [10–12]. Although the LLS spacing was sample-dependent, approximately the same spacing was observed. In the mammalian SC, the SLS spacing is about 6 nm. As will be explained later, by incorporating an aqueous layer between adjacent bilayers as shown in Figure 2, the spacing increased with the water content in the SC [7,12]. On the other hand, the ORTHO spacing is about 0.37 nm and about 0.41 nm [15–22], and the HEX spacing is about 0.41 nm [15,23,24]. Doucet et al. [16] by high resolution X-ray diffraction experiment revealed that the 0.41 nm peak was split into two peaks, consisting of 0.412 nm and 0.414 nm. Then, it is important to clarify which peak is the HEX peak and which is the ORTHO. Here, it should be pointed out that attenuated total reflectance Fourier spectroscopy (ATR-FTIR) is one useful tool that can detect the ORTHO and the HEX spectra separately and is possible to be measured in in vivo or tape-stripped SC sample [23]. These are not possible with X-ray diffraction. ATR-FTIR is therefore a complementary method to X-ray diffraction in structural studies of the SC. The broad peak of the LIQUID appears at about 0.45 nm [16]. This lies at almost same position to the peak of soft keratin as discussed later. In the intercellular lipids, the major lipid classes in the human SC are ceramides (CERs), cholesterol (CHOL) and free fatty acids (FFAs). The ratio of these lipid classes is approximately equimolar. In the human SC, more than 12 classes of CERs are present, where each one of these subclasses shows a variation according to their carbon number. FFAs show a variation in hydrocarbon-chain length. As a result, the total number of the intercellular lipids exceeds 350. Therefore, the intercellular lipid matrix is composed of a complex mixture of these lipids. Nevertheless, it is surprising that well-organized structures, such as the LLS, the SLS, the ORTHO, and the HEX, appear. This fact suggests that the individual structure has its own role in functioning in the SC.

The SC structure is formed mainly by three SC lipid classes, including ceramides, free fatty acids and cholesterol. Considering the size of these molecules is useful in gaining a deeper understanding of the structural arrangement of the SC, which is shown schematically in Figures 2 and 3. According to LC/MS analysis of all these SC lipid classes in the human SC performed, van Smeden et al. [24] reported that the relative abundance of FFAs is highest at the chain length of carbon number 24, where the chain length distribution spreads over carbon number from 12 to 36 and the relative abundance of CERs is highest at carbon number 46. Masukawa et al. [25] reported that the carbon number of the acyl chain in the CERs varies between 14 and 32, whose central value is about 23, whereas that of the sphingoid base varies between 14 to 28, whose central value is about 21. The relatively small size of the headgroups of free fatty acids and ceramides allows us to roughly estimate the molecular length from the number of carbons in each class. Since the distance per a

carbon atom in saturated hydrocarbon chain is 0.125 nm (in double bond of unsaturated hydrocarbon chain, 0.133 nm), in the FFAs the average length is approximately 3.0 nm and in the CERs the average length of the acyl chains is approximately 2.9 nm and that in the sphingoid base is approximately 2.6 nm. Although the level of the CERs (EO) subclasses is relatively low, as to the sphingoid base ω-hydroxy fatty acid is esterified and the carbon number is added. Therefore, the total the carbon number is roughly 48 [24]. In this case, the average length in the sphingoid base of the CERs (EO) is estimated to be approximately 6.0 nm. Therefore, it has been pointed out that in the formation of the long-period lamellar structure these molecules play considerably important contribution.

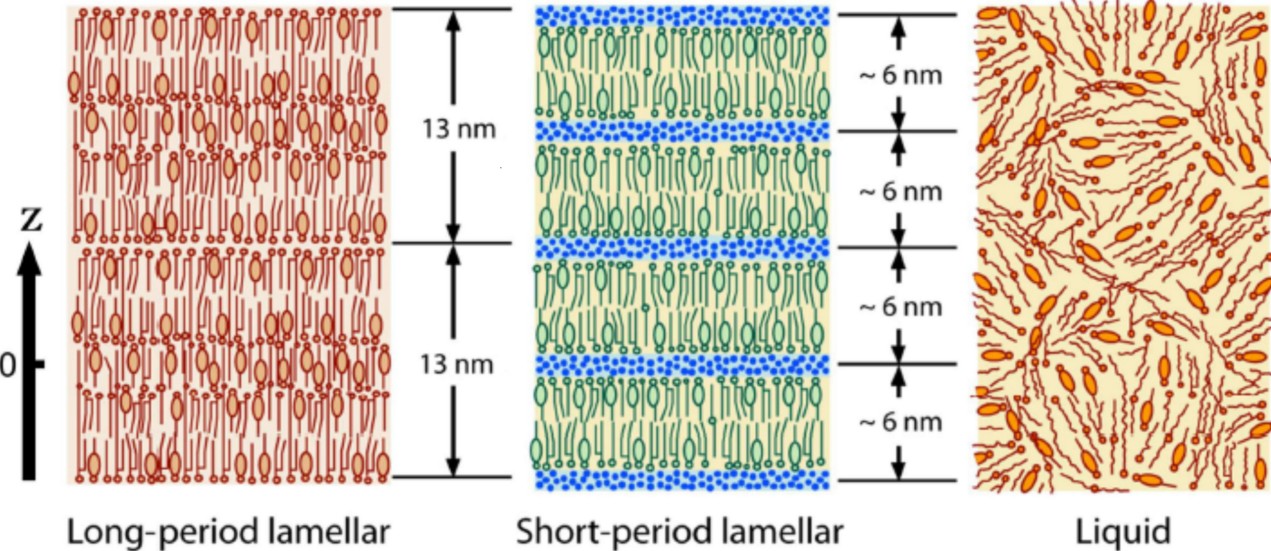

**Figure 2.** Schematic of the lamellar structures formed by the intercellular lipids; long-period lamellar structure (LLS); short-period lamellar structure (SLS); disordered liquid state (LIQUID). The molecular arrangement of ceramides, free fatty acids and cholesterol is drawn in them; in the narrow layer of the short-period lamellar structure, water molecules are inserted by small blue dots.

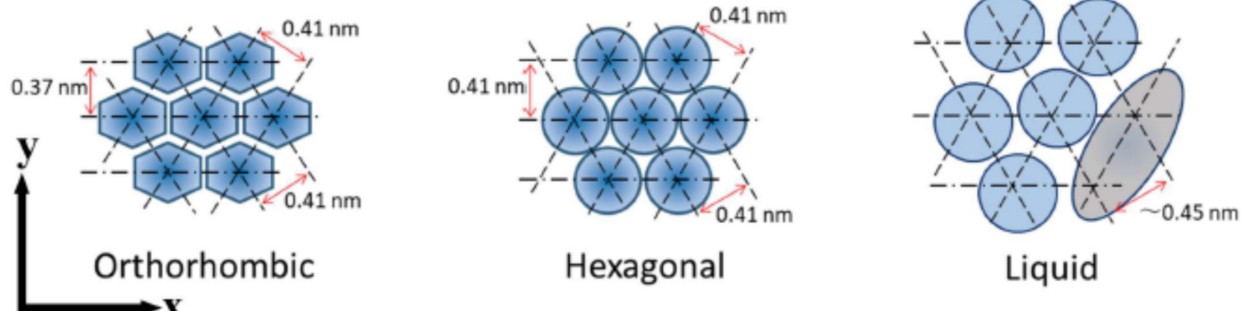

**Figure 3.** Schematic of hydrocarbon-chain packing structures formed by the intercellular lipids. In the cross-section perpendicular to the hydrocarbon-chain direction, the hydrocarbon-chain packing arrangement in orthorhombic and hexagonal hydrocarbon-chain packing structures (ORTHO and HEX, respectively) and liquid state (LIQUID) is shown. As shown in the structure of the liquid state, the molecular arrangement is greatly disturbed by the cholesterol molecule represented by the gray ellipse.

In the hydrocarbon-chain packing structures, there are the ORTHO, the HEX and the LIQUID. In the ORTHO, from the lattice constants of 0.41 and 0.37 nm we can estimated the distance between the hydrocarbon chains, that is, the first neighbor distance is about 0.44 nm and the second is 0.49 nm. This fact indicates that the motion around the long axis of the hydrocarbon chain is hindered. In the HEX, from the lattice constant of 0.41 nm,

there are two equivalent nearest neighbor distances of 0.48 nm. This fact indicates that hydrocarbon chains can rotate around their long axes.

In the X-ray diffraction of the SC, a peak due to a cholesterol monohydrate crystal was sometimes observed at $d$~3.35 nm. Craven [26] analyzed the crystal structure of cholesterol monohydrate. The structure shows a stacking of bilayers with a thickness of 3.39 nm, with the planes of the rings of a cholesterol molecule almost parallel to a plane on the long axis. Mojumdar et al. [27] performed neutron diffraction experiments in an SC lipid model. The position of a cholesterol molecule in the long-period lamellar structure was analyzed. They found that a cholesterol molecule lying parallel to the hydrocarbon chains has total length of ~1.6 nm, which is close to 3.39/2 nm. Regarding the hydrocarbon-chain packing structure, cholesterol distorts the arrangement of the hydrocarbon chains, as shown schematically in Liquid in Figure 3, due to the rather large ring width of the cholesterol molecule [26]. Therefore, a periodic appearance of cholesterol would be required to achieve a regular arrangement containing cholesterol molecules.

As can be seen in Figure 4, the corneocytes are shaped by roughly hexagonal-like plates having rounded edges with an average thickness of 1 μm and an average surface area of approximately 700 μm² [6]. Within the corneocytes, keratin filaments are oriented approximately parallel to the plane of the corneocytes, which almost coincides with the skin surface (see Figures 1 and 4B). Under normal conditions, almost all water in the SC is stored in the corneocytes by its interaction between soft keratin and natural moisturizing factors (NMF) [28,29]. By removing lipids in the SC with chloroform: methanol (2:1) a broad peak has been observed at about 0.45 nm and about 1 nm in the X-ray diffraction experiments [30]. Many researchers have reported that these are attributed to soft keratin in the corneocyte [30–34]. Based upon the above evidence, Kreplak et al. [34] summarized that the 1-nm broad peak corresponds to inter coiled-coil α-helix distance and the 0.45-nm broad peak is characteristic of disordered polypeptide chains. Hey et al. [35] have observed by the X-ray diffraction experiment on human callus that as a function of relative vapor pressure (r. v. p.) the 0.45-nm spacing was independent of r. v. p., but the 1-nm spacing increased with increasing r. v. p. and they proposed that the latter is principally due to interatomic distance between adjacent protein chains and consequently gives an approximate value of interchain separation. Following this result, Nakazawa et al. [36] performed the X-ray diffraction on the human SC. They found that the 1-nm spacing increased with increasing water content in the SC and nearly saturated at the water content of about 25 wt% but increased slightly above this water content.

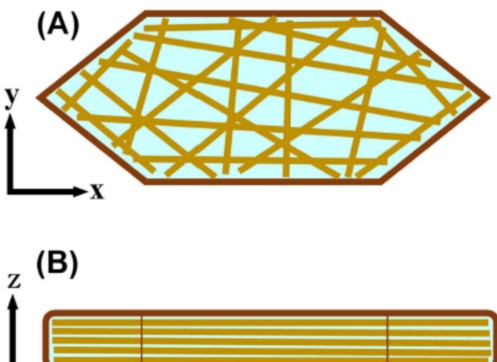

**Figure 4.** Corneocyte as illustrated as bricks in Figure 1. The corneocyte is a dead cell filled with soft keratin. The soft keratin filaments are arranged horizontally (*x-y* plane). (**A**) Top view; (**B**) side view.

The observation of the above structures was carried out by setting up the X-ray diffraction measuring apparatus as shown in Figure 5. For instance, samples of the unoriented SC were used for the measurements. In Figure 6, results are shown in the range from the small- to the wide-angle, that is, from small-scattering vector $q$ to large-scattering vector $q$.

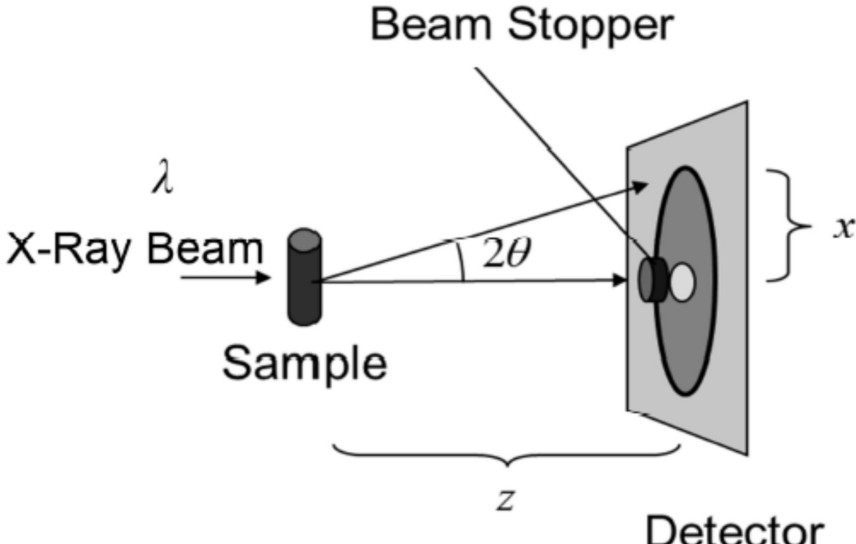

**Figure 5.** X-ray diffraction measurement system. To protect the detector from a strong direct X-ray beam, a beam stopper is set in front of the detector. Hence, in the smallest-angle region the X-ray diffraction profile is masked by the beam stopper.

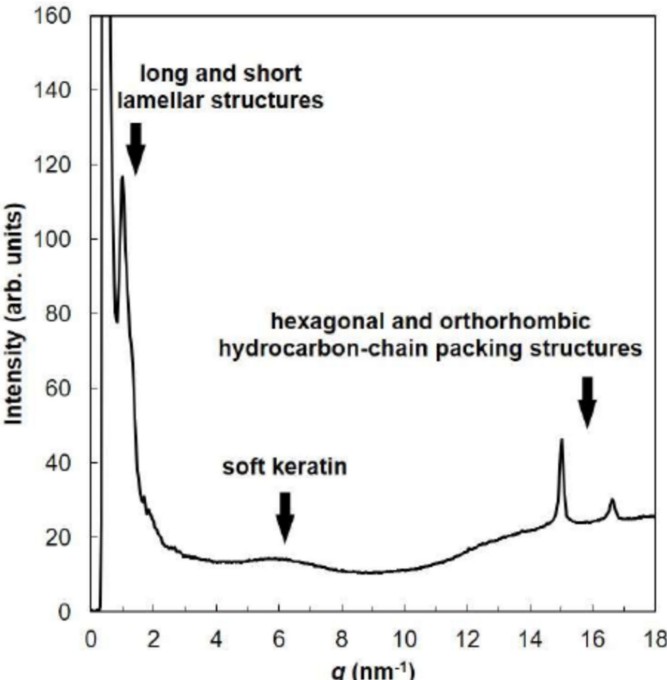

**Figure 6.** A typical X-ray diffraction profile of the stratum corneum from small- to wide-angle region, where lamellar structures, soft keratin structures and hydrocarbon-chain packing structures appear.

In the SC, the other anomalies have been observed in the X-ray scattering experiments. As seen in Figure 6, the smaller the scattering vector $q$, the greater the intensity. By the small angle neutron scattering experiments [37], which gives a complemental result with the small-angle X-ray scattering experiment, it has been observed that even smaller $q$ than $0.04$ nm$^{-1}$ continues to increase in the intensity as $q$ decreases. The result has been analyzed by an autocorrelation function, which gives a direct measure of the correlations at large scales in the SC. This may be because SC is roughly constructed by a stack of the corneocytes with a thickness of about 1 μm and a lateral size of about 30 μm and the intercellular lipid matrix that separates about 0.05 μm between corneocytes [37]. In analysis of the small angle X-ray diffraction experiments, it has been shown that the background intensity continuously

increases with decreasing $q$ from 1.5 to 0.2 nm$^{-1}$ [16]. Doucet et al. [16] have shown that the background scattering can be approximated by a power function of scattering vector as $I$(background)$\sim q^{-3}$ and in the analysis of the diffraction peaks the background intensity is subtracted; this subtraction procedure is robust, that is, small modifications of the power function do not lead to any significant modifications of the position or intensity of the peaks. There are many mechanisms that cause peaks toward $q = 0$ in the small-angle X-ray scattering. For instance, when the isolated structures composed of two or three units of tri-lamellar (single unit of the long-period lamellar structure) exist in the intercellular lipid matrix, it has been shown that the scattering intensity becomes significantly big in $q < 0.5$ nm$^{-1}$ as pointed out the experimental result and its analysis on the SC of not-previously-frozen human skin [38]. Another significant contribution to the X-ray scattering profile appears at high $q$. This comes from water. With increasing water content of the SC, a broad peak due to water is revealed at $q\sim 19.4$ nm$^{-1}$ (corresponding to $d = 0.324$ nm) [39]. Owing to the growth of the broad peak, the ascending gradient of the intensity profile at $q\sim 17$ nm$^{-1}$ changes from negative to positive with increasing the water content from the dried SC (see Figure 6). In addition, diffraction peaks of cholesterol crystals, neutral lipids, another orthorhombic hydrocarbon-chain packing, etc., have been observed in some cases. The weak peak at $q = 1.88$ nm$^{-1}$ ($S = 0.30$ nm$^{-1}$; $d = 3.35$ nm) is due to the presence of cholesterol crystals [11]. Cholesterol may precipitate in the SC because the intercellular lipids contain high levels of cholesterol. A relatively sharp peak appears at $q = 1.5$ nm$^{-1}$ ($S = 0.24$ nm$^{-1}$; $d = 4.1$ nm) due to the presence of neutral lipids such as glycerides [40]. These peaks disappear with rising temperature at about 70 °C, that is close to the transition temperature from the high-temperature hexagonal hydrocarbon-chain packing structure to the liquid state. In the wide-angle X-ray diffraction, in addition to 0.41- and 0.37-nm peaks a small peak appears at $q = 1.6$ nm$^{-1}$ ($S = 2.6$ nm$^{-1}$; $d = 0.38$ nm). This result is consistent with the orthorhombic hydrocarbon-chain packing structure with the spacings 0.42 nm and 0.39 nm obtained by the low-flux electron diffraction [22], where six Bragg's spots have been observed since the electron bean is very narrow and the observation is possible for a single domain SC sample. This fact indicates that in the hydrocarbon-chain packing structures not only the usual HEX and ORTHO, but other structures also exist.

As complementary methods to the X-ray diffraction experiments, for instance, confocal Raman microscopy is a powerful method to observe the water content and NMF concentration, as a function of depth from the skin surface [2]; transepidermal water loss (TEWL) is a useful tool for checking skin damage and, for instance, can be used to assess recovery from skin damaged by skin irritation with sodium lauryl sulfate [41]; electrical conductance or capacitance measurements are also conventionally a useful method for assessing average skin hydration [41], etc. Without this knowledge, it is impossible to have a deep understanding of what is going on in the SC. To understand the phenomena deeply, it is essential to elucidate the role of the structures by X-ray diffraction at the molecular level under as common conditions as possible. Here, the application of X-ray diffraction to the SC will be described.

## 2. Structural Study on the Stratum Corneum by X-ray Diffraction

### 2.1. What Can We Get from the X-ray Diffraction Experiments on the Stratum Corneum?

As an example of the X-ray diffraction experiments on the SC, I will describe an unoriented sample in which the SC is randomly filled in a sample holder made up of thin glass capillary tubes, as shown in Figure 5. An incident X-ray beam of wavelength $\lambda$, whose wave vector is given by $k_i$ ($|k_i| = 1/\lambda$), hits the sample and scatters in the direction of the scattering angle $2\theta$. The diffracted X-ray photons, whose wave vector is $k_d$ ($|k_d| = 1/\lambda$), are counted by a detector. A beam stopper is placed in front of the detector to prevent the strong beam from hitting the detector directly. As the SC sample was packed randomly, the diffracted X-ray intensities are distributed in an annular shape on the detector. The scattered X-ray intensity is circular-averaged to obtain the intensity profile as a function of scattering vector $q$ ($= |2\pi(k_d - k_i)|$) that is given by $q = (4\pi/\lambda)\sin(2\theta/2)$. A typical intensity

profile of the SC is shown in Figure 6 in the $q$ range from 0.3 nm$^{-1}$ to 18 nm$^{-1}$ where below 0.3 nm$^{-1}$ the X-ray scattering beam is masked by a beam stopper (see Figure 5). The diffraction peaks originated from the LLS and the SLS (see Figure 2) appear in the small-angle region, and the diffraction peaks originated from the HEX and the ORTHO (see Figure 3) appear in the wide-angle region. A broad peak associated with the soft keratin appears in the medium region (see Figure 3). These structures are modified by applying substances, such as pharmaceuticals and cosmetics, as described below.

### 2.2. Principle of X-ray Diffraction Experiments: With Attention to the Structures That Appear in the Stratum Corneum

Information on the electron density distribution in the SC can be obtained from the X-ray diffraction (XRD) experiments. In the X-ray diffraction experiments, the X-ray is scattered by electrons of atomic shells in a substance. Therefore, from the measurement of the X-ray diffracted intensity, we can derive the molecular arrangement in the substance. We give diffracted intensity, $I(S)$, and the electron density, $\rho(r)$, where $S$ is the scattering vector whose modulus is defined by $S (= q/2\pi) = (2/\lambda)\sin(2\theta/2)$, and $r$ is the vector defining the position in the substance. The Fourier transform of the electron density is given by structure factor $F(S)$,

$$F(S) = \int \rho(r) \exp(2\pi i r \cdot S) dr \tag{1}$$

The diffracted intensity is related to the structure factor $F(S)$ as

$$I(S) = KF(S)F^*(S) = K|F(S)|^2 \tag{2}$$

where $K$ is a correction factor and $F^*(S)$ is the complex conjugate of $F(S)$. The structure factor is given by structure amplitude, $|F(S)|$, and phase, $\varphi(S)$, as

$$F(S) = |F(S)| \exp(i\varphi(S)) \tag{3}$$

Once the phases are determined, the electron density distribution can be reconstructed from the structure factor by an inverse Fourier transform,

$$\rho(r) = \int F(S) \exp(-2\pi i r \cdot S) dS \tag{4}$$

In the X-ray diffraction measurement, we can obtain the diffracted intensity, for instance, from the LLS in the SC. However, in this measurement, information regarding the phase is lost. In the structural analysis of the LLS, we first consider the electron density distribution that changes one-dimensionally in $z$ direction in long-period lamellar of Figure 2, so the vectors $S$ and $r$ can be replaced with the scalars $S$ and $r$ (or $z$ in Figure 2).

In the LLS, a periodic structure with a repetition distance of about 13 nm appears. The structure is composed of a variety of molecules such as various fatty acids, many ceramides and cholesterol, nevertheless the repeat distance obtained by X-ray diffraction is surprisingly almost similar. As observed by electron microscopy, the basic molecular arrangement within a single lamellar structure with the repeat distance $d\sim$13 nm has a tri-lamellar structure [8]. The multilayer structure is formed by stacking the tri-lamellar structures. Since we want to know arrangement of the molecules in the LLS, it is important to analyze the electron density distribution which reflects the molecular arrangement. In Figure 2 the molecular arrangement of the LLS with the period, $d$, is drawn only schematically. In order to know the actual molecular arrangement, it is indispensable to obtain detailed knowledge of the electron density distribution. Referring to the tentative LLS shown in Figure 2, I will explain how to analyze the one-dimensional electron density distribution in a unit laying in $-\frac{d}{2} < z < \frac{d}{2}$. The electron density distribution, $\rho(z)$, is

given by the projection of the total electron density around the molecules in the LLS on the $z$-plane of Figure 2. Then, the structure factor of Equation (1) is rewritten as follows:

$$F(S) = \sum_{n=1}^{N} \exp(2\pi iSnd) \int_{-\frac{d}{2}}^{\frac{d}{2}} \rho(z) \exp(2\pi iSz)dz \equiv G(S)F_{unit\ cell}(S) \tag{5}$$

where $n$ is the $n$-th unit number and $N$ is the maximum unit number which contributes to X-ray diffraction. Since Laue function $|G(S)|^2$ is derived from $G(S)$ and $F_{unit}(S)$ is the structure factor for the unit, the intensity is derived from Equation (2) as

$$I(S) = K|G(S)|^2|F_{unit}(S)|^2 \tag{6}$$

When $N$ is sufficiently big, the peaks in the Laue function $|G(S)|^2$ show maximum at

$$Sd = h \tag{7}$$

where $h$ is integer (1, 2, 3, ... ). The maximum value becomes $N^2$ and the width $1/N$. Equation (7) is called the Laue condition and corresponds to the Bragg's condition given by the following equation.

$$2dsin\left(\frac{2\theta_B}{2}\right) = h\lambda \tag{8}$$

At a Bragg reflection: $q_B = (4\pi/\lambda)\sin(2\theta_B/2)$, the diffraction profile shows peaks. If there is a periodic structure, the Bragg reflections can be detected. Then, without knowing the electron density distribution, structural changes in the SC due to the application of an effective substance have been frequently discussed based only on the period $d$ obtained from Bragg reflections. Since some of the typical diffraction peaks are relatively sharp and strong, the peak changes due to the application are useful to study the modification of the LLS. However, knowledge of the electron density distribution is important to know the molecular interaction in detail.

From the observed intensity $I(h)$ ($h$ = 1, 2, 3, ... ), the electron density distribution is calculated by the following relation:

$$\rho(z) = \left(\frac{2}{d}\right) \sum_{h=1} \sqrt{I(h)h^2} \exp(i\varphi(h)) \cos\left(\frac{2\pi ihz}{d}\right) \tag{9}$$

Since the SC sample consists of a randomly oriented stack, in Equation (9), the intensity is corrected by multiplying the Lorentz factor $S^2$ (or $q^2$, or $h^2$ when sharp diffraction peaks appear.) as a correction factor $K$. Here, in the SC sample, the number of the molecules concerned with the X-ray diffraction experiment cannot be specified, so the unit of the electron density is given in arbitrary units. In the case of the oriented lamellar structure, the intensity should be multiplied by $S$ (or $q$) for the Lorentz correction.

### 2.3. Structural Study by X-ray Diffraction to the Further Analysis of the Long-Period Lamellar Structure

To determine the electron density distribution in the LLS in the SC is one of important purposes of the X-ray diffraction experiments. As seen in Equation (9), phase $\varphi(h)$ is required to obtain $\rho(z)$. In Equation (2), we can derive $|F(S)|$ from $I(S)$. For centrosymmetric systems where there is a center of symmetry in the middle of a unit as in the LLS (see Figure 2), $\varphi(S) = 0$ or $\pi$, that is, $\exp(i\varphi(S)) = +1\ or\ -1$ [42]. Therefore, the phase problem is reduced to a sign problem in this case. For each Bragg reflection, $F(S)$ can be positive or negative. In the procedure of selecting the correct phase for each Bragg reflection, Shannon's sampling theorem [43,44] is used to construct a continuous Fourier transform through one set of diffraction data. For any one set of data with the correct phase combination, there is ambiguity that the exact opposite phase combination gives a continuous Fourier transform with the same absolute values.

To obtain a continuous Fourier transform, the swelling method provides a reliable set. In this method, the positions of the Bragg reflections are varied by changing the lattice constant without changing the structure factor for the unit: $F_{\text{unit}}(S)$. For this, water is frequently used since the electron density of water is constant. To change the lattice constant, the width of the water layer between the adjacent layers is changed, so it is called the swelling method. If the width of the water is changed widely by this procedure, the structure amplitude changes along the continuous Fourier transform of the structure factor of the unit.

McIntosh [45] has performed the swelling experiment in a SC lipid model, which contains 2: 1: 1/ceramides: palmitic acid: cholesterol, where ceramides are composed of the major porcine ceramides. In this system, unlike the LLS the lamellar repeating distance increased from 12.1 to 13.3 nm by incorporating water in the aqueous layer. From the observed intensities of the diffraction peaks up to $h = 3$, the absolute values of the structure amplitude could be obtained. One set of data fits into the Fourier transform structure factor distribution in the eight phase selections, that is, $2^3$. From the plot of structure amplitude vs. scattering vector, it was found that the structure amplitudes for all of the data fall closely to the continuous transform calculated for the combination $(-, +, +)$ or $(+, -, -)$, indicating that one of these combinations is the correct phase choice. From a possible electron density distribution expected from the molecular arrangement in a lipid system with water layer, it was determined that the phase combination in this system is $(-, +, +)$ [45]. As a result, the electron density distribution can be obtained in this system from Equation (9). The original lamella repeating period in the SC is close to 13 nm, but swelling caused by the uptake of water appears in the LLS until no significant swelling is observed [11,12].

Groen et al. [13] have tried to obtain the electron density distribution in the LLS. They prepared various lipid mixtures composed of the isolated or synthetic ceramides, free fatty acids and cholesterol without water. In the analysis of the SAXD profiles observed for each mixture, six distinct diffraction peaks were used. In these mixtures, the peaks were attributed to the LLS with repeat distances in the range 12.1–13.8 nm. Structure amplitudes, $|F(S)|$, was calculated from the intensities of six diffraction peaks. A continuous Fourier transform function was calculated to fit the six structure factor, $F(S)$. Finally, a set of sign combinations was selected to be $(-, +, +, -, -, +)$. The first three signs are consistent with the result of McIntosh obtained from the 1st to 3rd order diffraction [45]. The signs or phases were determined consistently and extended up to 6th order diffraction. Since Groen et al. [13] did not employ water-induced swelling, it is not clear what kind of swelling mechanism occurs, but it is very attractive because the six data points are well represented by a continuous function.

In connection with the above suggestive results, it is important to recall the basic concept of the so-called swelling method. The swelling method without periodic appearance of a water layer has been proposed, in which the lattice constant can be varied in the interdigitated structures which are formed by dispersing phosphatidylcholine in alcohol [46] and in phosphatidylcholine in a glycerol solution [47]. In both studies, by increasing hydrocarbon chain length of saturated phosphatidylcholine, the lattice constant increased in the longitudinal direction parallel to the hydrocarbon chain, while the hydrocarbon chains were packed densely and uniformly in the lateral direction. Therefore, instead of the water layer, a band composed of hydrocarbon chain packing was considered, that is, the change of the longitudinal hydrocarbon chain length corresponds to the change of water layer thickness. In the latter [47], five saturated diacylphosphatidylcholines, namely dimyristoylphosphatidylcholine (C14PC), dipentadecanoylphosphatidylcholine (C15PC), dipalmitoylphosphatidylcholine (C16PC), diheptadecanoylphosphatidylcholine (C17PC), and distearoylphosphatidylcholine (C18PC), were used in the experiment. The interdigitated structure of these diacylphosphatidylcholines was formed by applying the glycerol solution at 10 °C. The lattice constant increased with increasing carbon number of saturated diacylphosphatidylcholines. From the structure amplitude vs. scattering vector plot, the continuous Fourier transform function was calculated using Shannon's

sampling theorem. Using the phases determined by this procedure, the electron density distribution was obtained. This shows that the moiety around polar headgroup regions remains almost unchanged, being independent of the carbon number, while outside this region the thickness of the hydrocarbon-chain region increases with the carbon number and the electron density of this region is nearly constant. The above result suggests that to solve the phase problem in a lipid system, it is possible to use the change of the carbon number of the lipid. Finally, we can obtain the electron density distribution in the digitated structure. From this point of view, it would be interesting if we could further consider the structure of the LLS.

I will briefly mention another possibility to determine the phase of the LLS. A hydrophobic chemical agent, *d*-limonene, is well known as an enhancer for percutaneous absorption of drugs [48]. By the X-ray diffraction experiments, it has been observed that, when *d*-limonene was applied to the hairless mouse SC, the spacing of the LLS increased [49]. This fact indicates that *d*-limonene penetrates the hydrophobic disordered banded region in the LLS, and as a result, the LLS expands. Therefore, *d*-limonene might be applied to the phase analysis. However, from the small angle X-ray diffraction experiments it has been pointed out that the excess application of *d*-limonene to the SC causes disruption of the LLS [33,50]. Based on these results, it is inferred that *d*-limonene is not a proper substance for the phase analysis, instead hydrophobic small molecules such as terpen except for *d*-limonene may be a candidate for this purpose. In fact, nerolidol has been shown not to seriously damage LLS. [33].

On the other hand, without knowledge of the phase the electron density distribution of the LLS in the SC was estimated by a strip model. Generally, to obtain the distribution of the LLS with high resolution, the intensity up to higher order diffraction peaks is required. From this point of view, the heat treatment established by Bouwstra et al. [11] is very interesting, where in the human SC, after heating 120 °C and cooling to room temperature, the seventh order diffraction peak was observed. They estimated that the mean number of the stack of the tri-lamellar structure *<N>* was about 6 from the width of the diffraction peak. From the distinct intensities up to the sixth order diffraction, by comparing the intensities observed experimentally and the peak intensities predicted from an electron density distribution that was given a priori, Bouwstra et al. [11] obtained the electron density distribution of hydrocarbon-chain regions. This is consistent with the result obtained from the electron micrography by Swartzendruber et al. [8], where $RuO_4$ was used to stain lipid. From the electron micrograph, Swartzendruber, et al. [8] proposed a model of the arrangement of hydrocarbon chains with one narrow hydrocarbon chain region and two wide hydrocarbon chain regions, resulting in two different hydrocarbon regions. In hairless mouse SC, using a similar heat treatment, Bouwstra et al. [14] observed the X-ray peaks of the LLS over 10th order. They estimated that the mean number of the stack of the tri-lamellar structure *<N>* was about 4. Using the intensities, the electron density distribution of the LLS in the SC was estimated by a strip model with two hydrocarbon chain regions or with three low-electron-density regions. Therefore, it may be difficult to establish the electron density distribution within the LLS by X-ray diffraction experiments with the strip model alone.

Yagi et al. [38] analyzed the results of the human SC on skin, not previously frozen, obtained from the small-angle X-ray diffraction. In their experiment, a 6 μm X-ray beam passed through the top layer of the human skin was used for small-angle X-ray diffraction. The diffraction profile was very similar to that observed in the human SC before heat treatment [11]. They proposed that the observed diffraction profiles were expressed by sum of the diffraction profiles resulting from three electron density distribution patterns that were constructed based on the electron micrographs of Swartzendruber et al. [8]. One distribution pattern is composed of the simplest unit consisting of a central density minimum and two minima that were at 5.0 nm on either side of the center with the same depth. Another two distribution patterns are composed of two and three units. In these patterns the densities with all minimums were assumed to be Gaussians. It was

hypothesized that the observed X-ray diffraction profile could be explained by the weighted sum of the diffraction profiles obtained from the three distribution patterns. The profile thus obtained is similar to the observed profile with peaks at $q$ = 0.56, 1.04, 1.41 nm$^{-1}$ and reflected a high baseline between the latter two peaks. In the diffraction profiles of the two-unit and three-unit distribution patterns, the corresponding three peak positions do not exactly correspond. However, the sum of them is similar not only in the peak positions, but also in the intensities. This study suggests that a small number of oligo-tri-lamellar structures are distributed in the LLS, but until this study a structure consisting of stacks of a relatively small number of tri-lamellar structures has been proposed. In addition, in further study, it is highly desirable to consider the contribution of the SLS whose first-order diffraction peak lies near $q$ = 1.04 nm$^{-1}$.

Summarizing the above results, in the LLS, the number of stacks of the tri-lamellar structure, $N$, is relatively small, as obtained by the electron microscopy observations [10,11] and estimated from the X-ray diffraction experiments [11,38]. In the X-ray diffraction studies on the LLS of the SC, Yagi et al. [38] analyzed the results using Equation (4) with one-dimensional electron distribution of one- two- and three-unit distribution patterns. In the analyses by Bouwstra et al. [11] and Bouwstra et al. [14], the electron density distribution was assumed a priori. The structure factor was calculated by Equation (5), then the intensity was obtained from Equation (6). By repeating this procedure, the most appropriate electron density distribution could be searched. When the Laue function $|G(S)|^2$ is approximated by Gaussian function, we can introduce Scherrer's equation, and calculate <$N$>. In the above studies [13,14], the mean number of stacks of the tri-lamellar structure, that is obtained from Scherrer's equation, <$N$>, was about 6 or 4, from the width of the diffraction peak. In the structural analysis of the human SC, X-ray diffraction should consider that the stack number of the unit is relatively small.

The X-ray diffraction experiments are indispensable to elucidate the structure of the SC related to the function. To proceed with detailed structural research further, it is essential to refer to the structural studies, neutron diffraction, electron diffraction, electron microscopy, etc. and also to other indirect methods. The structural studies in stratum corneum lipid models are also appropriate if they may provide valuable evidence for functioning in the SC.

*2.4. X-ray Diffraction Studies on Short-Period Lamellar Structure, Soft Keratin, and Ordered Hydrocarbon-Chain Packing Structures*

In the study on the hydration effects on the SC, it is very important to clarify the changes in the SLS due to water, as swelling of the SLS appears by taking water in the SC [7,36]. As shown in Figure 7, in the hairless mouse SC, a swelling effect is clearly observable. In Figure 8a, the changes of the spacing of the SLS and half of the spacing of the LLS are shown as a function of the water content. It is clear that the SLS shows swelling but the LLS does not. In Figure 8b, the full width at the half maximum of the X-ray diffraction peaks is shown for the SLS and the LLS as a function of the water content. It is clear that the width becomes narrow at the water content of about 25 wt% in both SLS and LLS. This result indicates that not only the SLS, but also the LLS become a stable state at the water content of about 25 wt% [11,12,36]. However, in the human SC, the first-order X-ray diffraction peak of the SLS is strong but very wide, although the second-order diffraction peak is small [11,12]. From the analysis of the broad first-order X-ray diffraction peak of the human SC, it has been proven that the SLS shows similar behavior to that in the hairless mouse SC [36]. For the detailed analysis in the human SC, it is highly desirable to separate the first-order diffraction peak of the SLS and the second-order peak of the LLS appearing at $q \approx 1$ nm$^{-1}$. As pointed out before, the neutron diffraction experiment in the SC using heavy water is a powerful tool to overcome this problem [7], since there exist periodically water layers in the SLS. Since one of the dominant subjects in the SC is related to the behavior of water, to perform the structural study on the SLS under in various conditions of water is extremely important.

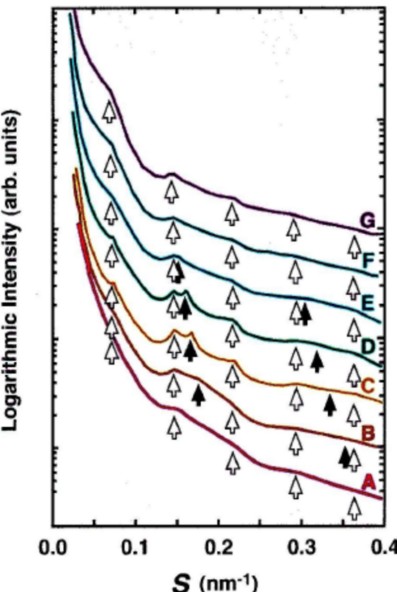

**Figure 7.** Semi-logarithmic profiles of small-angle X-ray diffraction intensity as a function of the scattering vector (*S*) in units of nm$^{-1}$ for the stratum corneum of hairless mouse at the water contents: (A) 0 wt%, (B) 12 wt%, (C) 21 wt%, (D) 35 wt%, (E) 50 wt%, (F) 70 wt% and (G) 80 wt%. Each profile is moved upward successively to see easily. Open arrows indicate the 1st to the 5th order diffraction peaks for the long-period lamellar structure and closed arrows indicate the 1st and 2nd order diffraction peaks for the short-period lamellar structure. Reprinted with permission from ref. [12] Copyright 2022 Elsevier.

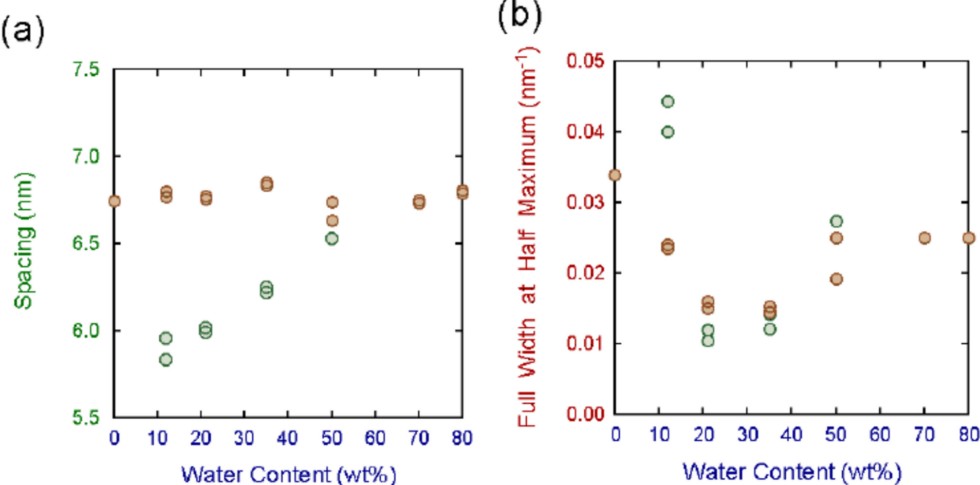

**Figure 8.** (**a**) Dependence of spacing on the water content obtained from the 2nd order diffraction of the long-period lamellar structure (brown dot) and from 1st order diffraction of the short-period lamellar structure (green dot). (**b**) Full width at half maximum of X-ray diffraction peaks for the 2nd order diffraction of the long-period lamellar structure (brown dot) and full width at half maximum for the 1st order diffraction of the short-period lamellar structure (green dot). Reprinted with permission from ref. [12] Copyright 2022 Elsevier.

Soft keratin in the SC shows characteristic X-ray diffraction profile; two strong maxima observed at the spacings of approximately 1.0 nm and 0.46 nm [16,32]. This 1-nm spacing seems to be related to that of approximately 1.0 nm in callus of skin [34]. A broad scattering maximum located at about 1.0 nm in callus is supposed to be due to interferences between coiled-coil α-helix chains [30,51,52]. However, Garson et al. [30] pointed out that the spacing of approximately 1.0 nm was slightly bigger in callus than that in the SC although in the

latter it depended on the sample preparation. Based upon the various physical properties of keratin Wang et al. [53] state that (i) α-keratin is found in mammals, (ii) the 1.0 nm spacing corresponds to the distance between α-helix chains. From them, it is inferred that the 1.0 nm spacing in the SC is due to the coiled-coil α-helix structure in soft keratin.

In the ordered hydrocarbon-chain packing structures, the ORTHO and the HEX, the ratio of the integrated intensities for 0.41 nm and 0.37 nm spacing, $I_{0.41}/I_{0.37}$, has been frequently discussed, since the ratio, $R$, may offer evidence of a population of the ORTHO and/or the HEX. At 0.41 nm peak, the peaks due to the ORTHO and the HEX are superposed, and therefore the intensity is given by $I_{0.41} = I_{0.41}{}^O + I_{0.41}{}^H$. On the other hand, the 0.37 nm peak is due to the ORTHO alone, and therefore $I_{0.37} = I_{0.37}{}^O$. The ratio is given by

$$R = \frac{I_{0.41}}{I_{0.37}} = \frac{I_{0.41}^O}{I_{0.37}^O} + \frac{I_{0.41}^H}{I_{0.37}^O} = R^O + \frac{I_{0.41}^H}{I_{0.37}^O}. \tag{10}$$

In Equation (10), the first term on the right-hand side, $R^O$, is constant even if the intensities change due to some action, because the ratio due to the ORTHO, originated from the basic structure, should be maintained. Then, we can deduce that the change of $R$ due to some action appears in the second term on the right-hand side. Here, many other diffraction peaks need to be considered since many hydrocarbon-chain packing structures appear, as mentioned previously. In relation to this fact, we should pay attention to the fact that the 0.37 nm peak exhibits asymmetric. In fact, according to the result of the hairless mouse SC [54], splitting of the 0.37 nm peak has been observed clearly (see Figure 2b of this reference [54]). In the analysis of the spacings and the intensities of the ordered hydrocarbon-chain packing structures, we should pay attention to such complexity.

### 3. Disruption and Reconstruction of Long-Period Lamellar Structure in Stratum Corneum

#### 3.1. Disruption of Human Stratum Corneum Lipid Structure by Sodium Dodecyl Sulfate

An anionic surfactant, sodium dodecyl sulfate (SDS), is used as an additive to cleansing-products. However, in the SC, treatment with concentrated SDS aqueous solution results in increase of TEWL [55,56]. In previous studies, it has been pointed out that SC lipids are extracted by surfactants [57,58], resulting in damage to the SC [59,60]. In addition, exposure to surfactants leads to changes in SC morphology such as swelling and exfoliation [61–64]. Although these studies are investigating cleansing-induced skin damage, most studies to date have not focused on structures formed in the intercellular lipids of the SC, so the detailed underlying mechanism at the molecular level has not yet been established.

Synchrotron X-ray diffraction was used to investigate structural change in the human SC resulting from sodium dodecyl sulfate (SDS) treatment, focusing on how SDS affects SC lipid structures such as the lamellar structures and the hydrocarbon-chain packing structures [65,66]. Here a special sample cell, called a "solution cell" developed by Hatta et al. [49], was used. A thin SC sample was crumpled and embedded in a hollow surrounded by a fine mesh composed of a glass microfilter. This fine mesh was used to sustain the SC sample, allowing solution flow through the mesh without clogging. As a result, the SC sample would be surrounded with full of solution. This method is particularly useful for detecting minute changes in structure. In general, applying solution to an SC sample causes a gradual change in the structure, and continuous observation of the XRD profile allows to detect the very delicate changes in a single SC sample. Using the solution cell, we were able to detect structural changes within seconds by synchrotron XRD. Owing to these characteristics, we can overcome the problem of individual differences among the SC samples. Although the XRD peak intensity depends on the SC sample due to individual differences, systematic change in the intensity occurs regardless of the individual samples, then it is possible to obtain characteristic structural changes in a single SC sample. This method was applied to studying on the structural change by applying SDS to the SC.

The experiments were performed at BL40B2 (Structural Biology II Beamline) of SPring-8 (Hyogo, Japan). The measurements were performed every 300 s at room temperature.

The exposure time of the X-ray beam for the SAXD and the WAXD was 30 s. Using the solution cell, when SDS aqueous solution was applied to the human SC, in the small-angle X-ray scattering region we could observe the profile shown in Figure 9. Panel A shows the results on applying 10 wt% SDS aqueous solution; B on applying 1 wt% SDS aqueous solution. The profiles indicated by "pre" in panels A and B are almost similar in shape, but slightly different from sample to sample due to individual difference, except for a peak appearing at $S = 0.24$ nm$^{-1}$ in panel B that comes from neutral lipids [40]. In "pre" profiles of panels A and B, wide peaks appeared at $S\sim0.15$ nm$^{-1}$ ($q\sim1.0$ nm$^{-1}$) and $S\sim0.22$ nm$^{-1}$ ($q\sim1.4$ nm$^{-1}$). The peak at $S\sim0.22$ nm$^{-1}$ denoted by a white arrow on each panel is assigned as the third-order diffraction peak of the LLS. The peak at $S\sim0.15$ nm$^{-1}$ is mainly derived from the first-order peak of the SLS [11]. At $S\sim0.29$ nm$^{-1}$, the fourth-order peak of the LLS appeared, although the peak intensity is faint. As seen in Figure 9, when the SDS aqueous solution was applied to the SC, the third-order LLS peaks at $S\sim0.22$ nm$^{-1}$ changed and finally disappeared; the fourth-order LLS peaks at $S\sim0.29$ nm$^{-1}$ seemed to disappear finally. On the other hand, the peak at $S\sim0.15$ nm$^{-1}$ did not change. These facts indicate that when the SC is treated with SDS aqueous solution, the LLS is significantly altered and eventually disappears, but the SLS is largely unaffected.

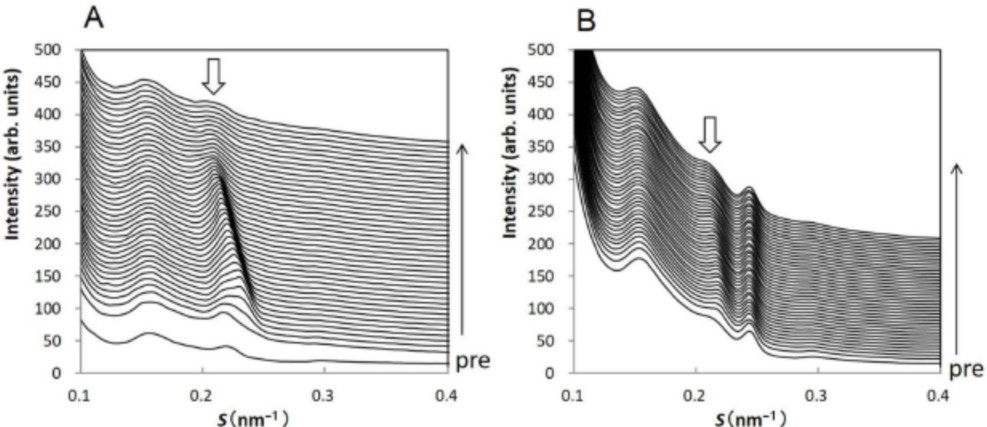

**Figure 9.** Small-angle X-ray diffraction profiles in human stratum corneum applying sodium-dodecyl-sulfate aqueous solution: (**A**). Treated with 10 wt% sodium dodecyl sulphate; (**B**). Treated with 1 wt% sodium dodecyl sulphate. The profiles before treatment are indicated by "pre". The profiles after the application were recorded every 3 min up to 129 min. The illustrated intensity profiles are shifted in the longitudinal direction. The arrow on the right side indicates the direction of the time course from 3 to 129 min. White arrows denote the third-order peak for the long-period lamellar structure. For a peak at 0.24 nm$^{-1}$ in B, see text. Reprinted with permission from ref. [65] Copyright 2022 John Wiley & Sons, Inc.

The profile of the third-order LLS peak at $S\sim0.22$ nm$^{-1}$ in Figure 9A was analyzed by fitting it with two Gaussian curves and a straight base line. In Figure 10A-I,A-II show the spacings and the integrated intensities obtained from the results in Figure 9A as a function of time, respectively, where the SC sample was treated with 10 wt% SDS aqueous solution. As seen in Figure 10A-II, the integrated intensity denoted by open circle is dominant and that denoted by closed circle is secondary, therefore the dominant integrated intensities first increase, then decrease and finally disappear, and the secondary one quickly decreases and disappears. As seen in Figure 10A-I, the spacing denoted by an open circle decreases quickly in the beginning, then gradually increases, while that denoted by closed circle monotonously increases. In the SC sample treated with 1 wt% SDS aqueous solution, the profile of the third-order LLS peak at $S\sim0.22$ nm$^{-1}$ in Figure 9B was analyzed by fitting it with a single Gaussian curve and a straight base line. As seen in Figure 10, the plots of B-I the spacing and B-II the integrated intensity are like the trends of the dominant ones in Figure 10A, but the changes appear slowly. Due to the weak intensity in question, the data

shown by the closed circles in Figure 10A-I,A-II could not be detected in the sample treated with 1 wt% SDS aqueous solution. Furthermore, the fourth-order diffraction peak intensity was very weak in Figure 9A,B, but this peak diminishes in accordance with the third-order diffraction peak. As can be seen in Figure 10A-I, the spacing of the dominant peak denoted by open circle drops rapidly at the beginning of treatment with the 10 wt% SDS aqueous solution. From the corresponding spacing in the treatment of the 1 wt% SDS aqueous solution, it was found that the spacing gradually decreased and increased. Therefore, the rapid decrease in the spacing in the treatment of the 10 wt% SDS aqueous solution is not unnatural. In the previous XRD studies of SDS-treated SC, it is pointed out that the lamellar structure is affected by SDS [40,67], but the detail of the structural changes was unclear. In our study [65,66], we found that SDS significantly alters the LLS in the SC.

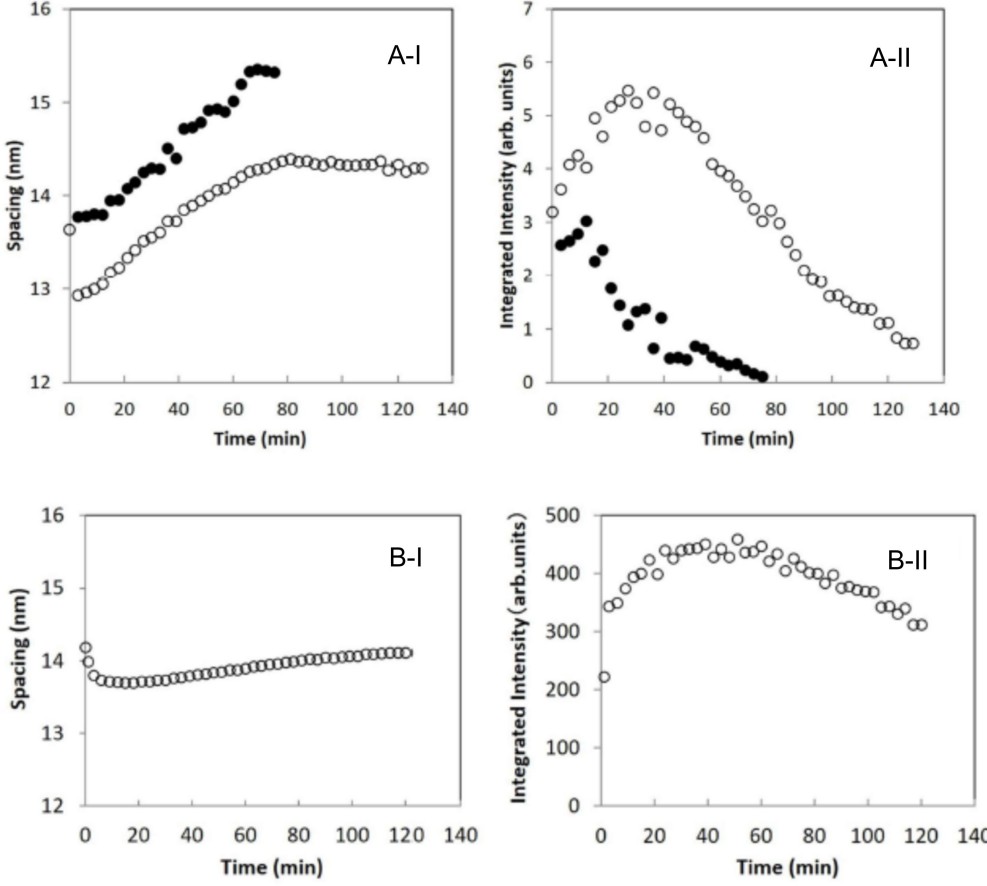

**Figure 10.** For the long-period lamellar structure of the SC treated with 10 wt% sodium dodecyl sulphate, (**A-I,A-II**) show time dependence of spacing and intensity of the third-order X-ray diffraction peaks, respectively. The dominant peak is indicated by open circles and the second peak is indicated by closed circles. Reprinted/Adapted from ref. [65]. For the long-period lamellar structure of the SC treated with 1 wt% sodium dodecyl sulphate, (**B-I,B-II**) show that of spacing and intensity of the third-order X-ray diffraction peaks, respectively. In this measurement, only the dominant peak was detectable.

The incorporation of SDS into the SC might be like the incorporation of cholesterol sulphate, which has the same sulphate moiety as SDS. According to an X-ray diffraction study on an SC lipid model [45], cholesterol sulphate molecules are located at either side of a repeating unit of about 13 nm, and the lamellar repeat distance is modified by taking up the cholesterol sulphate. In relation to this finding, a study by Downing et al. [68] on the incorporation of SDS into the SC is noteworthy. In an SC lipid model, they found that a high degree of partitioning of SDS molecules occurs. Therefore, it is plausible that many SDS molecules are incorporated into the intercellular lipid structure of the SC.

This is consistent with our finding that the LLS is markedly changed by SDS treatment. Bouwstra et al. [69,70] and Kuemple et al. [71] have pointed out that the LLS plays an important role in maintaining skin barrier function. In connection with this fact, it is important to study further the mechanism of structural modification of the LLS by SDS. Since the XRD peak of the LLS was split into two peaks by applying SDS to the SC, it suggests that there are two different crystallographic sites where SDS is incorporated in the LLS. It is worthwhile to consider the mechanism based on the result obtained from the neutron diffraction study on an SC lipid model. Mojumdar et al. [27] pointed out that water molecules localize in two crystallographic locations in the long-period lamellar structure, where one of them seems to be close to the location of cholesterol sulphate [45]. Based on this result, we propose the mechanism that SDS was incorporated in the two locations and, as a result, two types of structural change took place [66].

In relation to the above fact, it is interesting to note that, when water is applied to the SC, the X-ray diffraction profile of the LLS shows slight but characteristic alternation, that is, the width of the diffraction peak becomes narrow at about 25 wt% [11,12]. This might be associated with the water localization in the LLS of the SC. Therefore, it is deduced that water in the SC rules not only the water regulation, but also the barrier function.

### 3.2. Reconstruction of Damaged Long-Period Lamellar Structure

Decreased acylceramide contents in the SC have been associated with severe pathological consequences [72–77] and with various physiological effects on skin barrier function [78–80] and lead to a decrease in the proportion of the LLS [81,82]. In connection with this behavior, in SC lipid models, it has been shown that acylceramide is necessary to form the LLS [27,82–85]. If there is such damaged skin, topical application of acylceramide is considered an attractive approach to recover the barrier function of the damaged skin. Nakaune-Iijima et al. [86] demonstrated that the application of acylceramide, which is one of the important components in the LLS, to the damaged skin promotes the recovery of the LLS.

In the study by Nakaune-Iijima et al. [86], a human epidermal model (RHE) and corresponding culture media obtained from MatTek Co was used. The SC derived from the RHE sample was treated with 1 wt% SDS aqueous solution for 15 min, then the damaged SC was prepared, since by SDS treatment it is known that the LLS was disrupted [65,66]. To examine the recovery from the disrupted LLS, the acylceramide dispersion was applied to it for 8 h per day for 2 d. To obtain the acylceramide dispersion, a mixture was first prepared by mixing acylceramide with cholesterol and 1,3-butylene glycol. Secondly, an aqueous dispersion of lecithin was added to the mixture. Third the overall material was sonicated and emulsified. By dynamic light scattering analysis, the average diameter of nanoparticle formed in the dispersion was observed to be about 10 nm and remained stable throughout the experiments.

In Figure 11, the small-angle X-ray diffraction profiles are shown. The first profile from the bottom was observed in an untreated SC sample and the second one was observed in an SC sample disrupted by SDS. The third, fourth and fifth profile from the bottom were observed in an SC sample reconstructed by acylceramides, ceramide EOP-S, EOP-L and EOS-L, dispersion, respectively. In these profiles, the first-order X-ray diffraction peak is broad and appears at approximately $q = 0.52$ nm$^{-1}$ (spacing is ~12 nm). The peak intensities were analyzed by subtracting a baseline proportional to $q^{-\beta}$, where $\beta$ values were chosen so as to fit each profile outside the first-order X-ray diffraction peak. By analyzing the contribution of the first-order X-ray diffraction peak of the LLS, it was found that the integrated intensity of the peak was proportional to approximately 38; 1; 20; 30; 25 in the SC samples untreated; damaged by SDS; treated with ceramide EOP-S; EOP-L; and EOS-L, respectively. From this result, it was found that treatment with acylceramides can restore the damaged LLS.

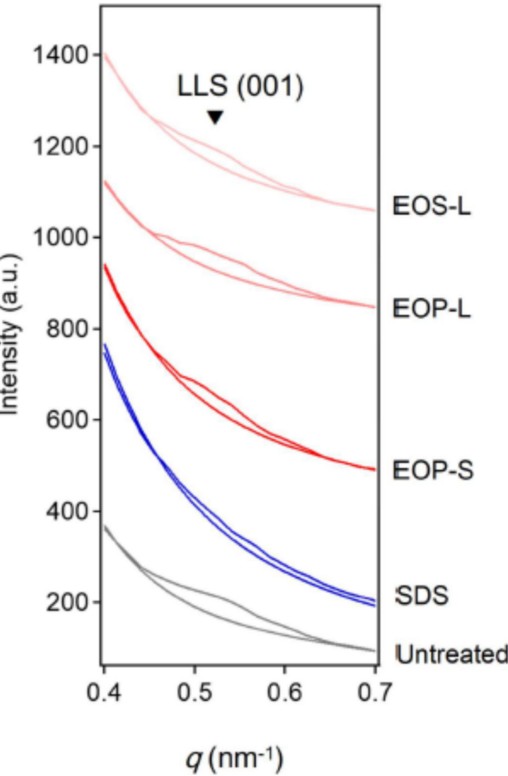

**Figure 11.** Small-angle X-ray diffraction profiles of the long-period lamellar structure (LLS) in the stratum corneum obtained from a human epidermal model, treated with water (gray), with sodium dodecyl sulfate following water treatment (blue), and with acylceramides EOP-S (red); EOP-L (light red); EOS-L (pink) following sodium-dodecyl-sulfate treatment. The baselines are inserted (see text) and ▼ indicates the first-order diffraction peak of the long-period lamellar structure. Adapted with permission from ref. [86] Copyright 2022 Elsevier.

Figure 12 shows a typical example of the electron micrographs observed in the corresponding SC sample compared to that observed by the small-angle X-ray diffraction experiments. SC samples stained by $RuO_4$ for the electron microscopy observation were prepared according to the method developed by Swartzendruber et al. [87]. The upper and lower micrographs in Figure 12a observed at the different positions are obtained for untreated SC samples, those of (b) damaged SC samples by SDS and those of (c) treated SC samples with acylceramides. As seen in Figure 12a, the electron micrographs of untreated SC samples are consistent with that obtained previously [8]. The micrographs shown in Figure 12b, and (c) are for the RHE SC sample damaged by SDS and the reconstructed ones by application of acylceramides, respectively. Figure 12d indicates the brightness distribution at the position marked in the lower micrograph of Figure 12c. The LLS period obtained from the brightness distribution obtained from the electron density distribution is at 12–14 nm. Therefore, the results are consistent with those obtained from the small-angle X-ray diffraction for the reconstructed SC sample, although the spacing seems to be a little different, probably due to the sample preparation.

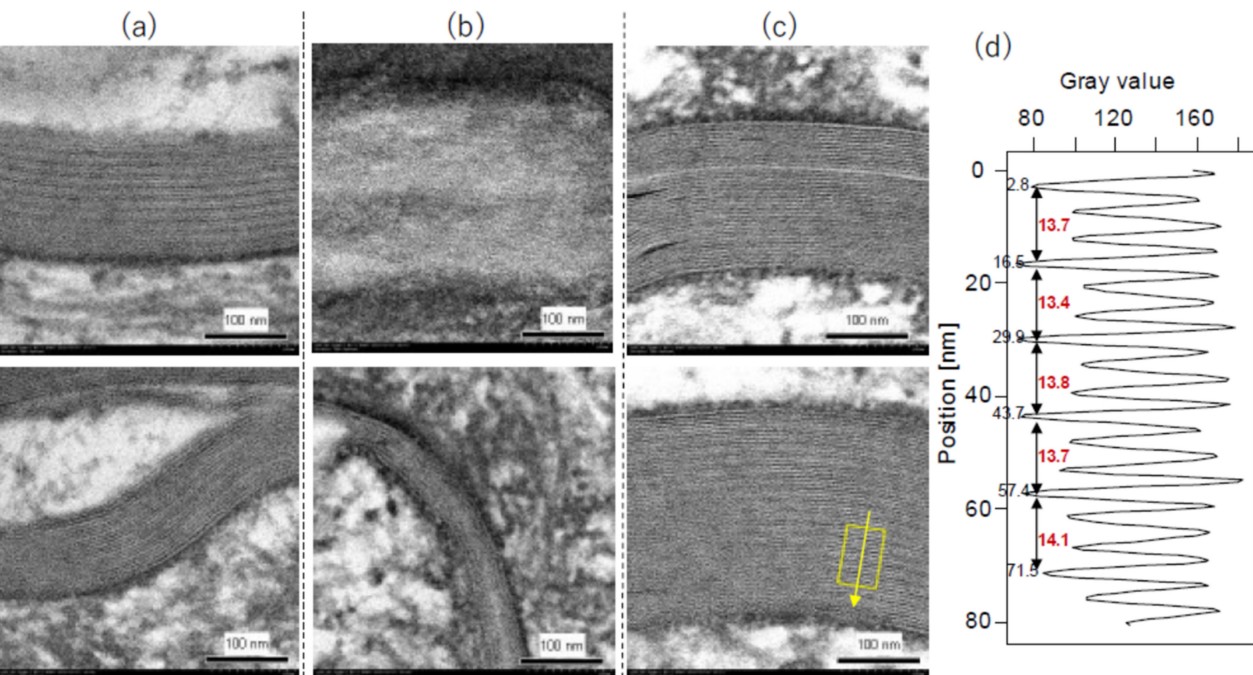

**Figure 12.** Transmission electron microscope images of the stratum corneum treated with (**a**) water, (**b**) with sodium dodecyl sulfate following water treatment, (**c**) with acylceramide dispersion following sodium dodecyl sulfate treatment. Brightness distribution analyses were performed in the image enclosed by the yellow box inserted in the lower image of (**c**). (**d**) Brightness distribution of the long-period lamellar structure. Reprinted with permission from ref. [86] Copyright 2022 Elsevier.

Some mechanisms for the recovery of the damaged LLS with acylceramide dispersions can be proposed. First, this is a clear simple forward mechanism that acylceramides directly infiltrate damaged LLS defects. Second, it is rather complicated but might be possible. This is a mechanism that acylceramides promote the new LLS formation in the intercellular lipid matrix. In connection with the second mechanism, it is worth considering the liquid state (LIQUID) in the intercellular lipid matrix, which is composed of ceramides, fatty acids and cholesterol [11,16–18]. It can be pointed out that the uptake of acylceramide into the LIQUID promotes nucleation and can generate the new LLS. This could be extended to the further idea that incorporating one of the components of ceramides, fatty acids, and cholesterol into the LIQUID promotes the formation of ordered states, such as the LLS and the SLS. In fact, it was found that, when an aqueous dispersion of the mixture of ceramide NP, cholesterol, and stearic acid with the molar ratio of 1:1:1was applied to the damaged SC treated with the SDS, the new LLS was formed [88].

## 4. Water Retention and Moisturizing in Stratum Corneum

### 4.1. Key Water Content 25 wt% in Stratum Corneum

Although normal skin surfaces are exposed to a variety of serious environments, their physiology and function are controlled to remain normal so that they can remain healthy. Occasionally, the surface may be exposed to a very dry condition. Even under such a condition, the water content of the skin surface must be kept unaffected seriously. Here, I consider the behavior of water in the outermost layer of skin that controls such a mechanism at the molecular level.

In vivo CRM experiments represent a powerful tool for detecting the distribution of the water content in the SC. Many researchers have published with almost comparable results [2–4]. The water content is low near the surface of the SC and increases from the surface to the viable cell, where it reaches approximately 70 wt%. When depth profile of the water content is linearly extrapolated toward the surface from the deep SC, the water

content at the surface is estimated to be close to 25 wt%. The in vivo CRM experiment by Egawa and Kajikawa [4] provides important evidence. At room temperature, they applied water to the skin surface, using a cotton wool pad, and found the increase of the water content near the surface. Then, at about 10 min after the water application, the depth profile of the water content returned almost to the initial depth profile. Consequently, we can regard that in the steady state, the water content near the surface is, in principle, around 25 wt% under the condition that water evaporates continuously from the surface through the SC.

Clear evidence indicating that 25 wt% is the key water content in skin has been obtained from differential scanning calorimetry (DSC). The transition enthalpy at the melting point of water at 0 °C has been measured as a function of the water content in the SC [89,90]. It has been shown that the transition enthalpy decreases as the water content decreases and that the transition enthalpy does not appear below the water content of 25 wt% (=((weight of water/(weight of dry SC + weight of water)) × 100 wt%)). In the definition of Imokawa et al. [90], this value corresponds to the water content 33 wt% ((=weight of water/weight of dry SC) × 100 wt%). This result indicates that if the water content is less than 25 wt%, there are water molecules tightly bound in the SC, and if the water content is greater than 25 wt%, excess water molecules above 25 wt% is free water. Combined with the results of the CRM experiments, it is inferred that water near the surface of the skin is mainly composed of tightly bound waters with the water content 25 wt%.

In addition to the overall properties mentioned above, it is very important to reveal role of the 25 wt% water content at the molecular level. As mentioned in Section 1, under normal conditions almost all water is stored in corneocytes. Water in corneocytes interacts with soft keratin via NMF, as a result the water is retained. Based on the general structure of keratin, Jokura et al. [28] proposed that for filaments in soft keratin they are substantially divided into two domains, the flexible segment and the rigid rod-like core. The long direction of the rigid core lies parallel to almost the surface of the plate-like corneocytes as shown in Figure 4B and the flexible segments are protruded from the rigid core. In Figure 13A, the arrangement of the rigid core and the flexible filaments is schematically shown (The detail interpretation of this figure is made later). Today, based on the knowledge of the X-ray diffraction studies, the structures observed in soft keratin reveal that an amorphous halo around 0.45 nm is characteristic of disordered polypeptide chains and a broad meridional X-ray diffraction spot at about 1 nm corresponding to the mean distance between two $\alpha$-helical chains [34]. The latter X-ray diffraction spot is insensitive to lipid extraction and is therefore attributed only to soft keratin. On the other hand, the former diffuse ring around 0.45 nm is sensitive to lipid extraction. The ring is composed of a superposition of the diffraction profiles from the secondary structure elements of keratin and from the liquid state of the extraction-sensitive intercellular lipids characterized by an average interchain distance of 0.45 nm.

From the cross-polarization/magic angle spinning $^{13}$C-nuclear magnetic resonance NMR) of aliphatic carbons in the non-helical flexible segment of soft keratin [28], it was found that under the presence of natural moisturizing factors (NMF), as the water content increased from the dry SC, the relative intensity of the nuclear magnetic resonance spectrum decreased significantly. Then, above the 25 wt% water content where free water is contributed the relative intensity decreased slowly. The result indicates that the initial decrease of the relative intensity is associated with accumulation of bound waters up to about 25 wt% and with gaining elasticity. Therefore, in the initial hydration process in the SC are mainly retained in the corneocytes as bound water and are accumulated around non-helical flexible filaments of soft keratin via NMF. When the water content exceeds 25 wt%, free waters enter the corneocytes so that the NMR relative intensity slowly decreases. In connection with the above relative intensity change in the non-helical flexible segment, it is important to pay attention to the correlation of the relative intensity changes between for aliphatic carbons in nonhelical flexible segment and carbonyl carbons in $\alpha$-helical structure in rigid core. With hydration from the dry SC, the relative intensity for the $\alpha$-helical structure also

changed according to the relative intensity for the nonhelical structure, although water does not directly affect the α-helical structure. Namely, up to the water content 25 wt%, in the rigid core, the relative intensity initially decreases in accordance with decrease of the relative intensity in the flexible segments that play a dominant role in gaining elastic properties and flexibility. Above 25 wt%, the relative intensity also slowly decreases in the rigid core.

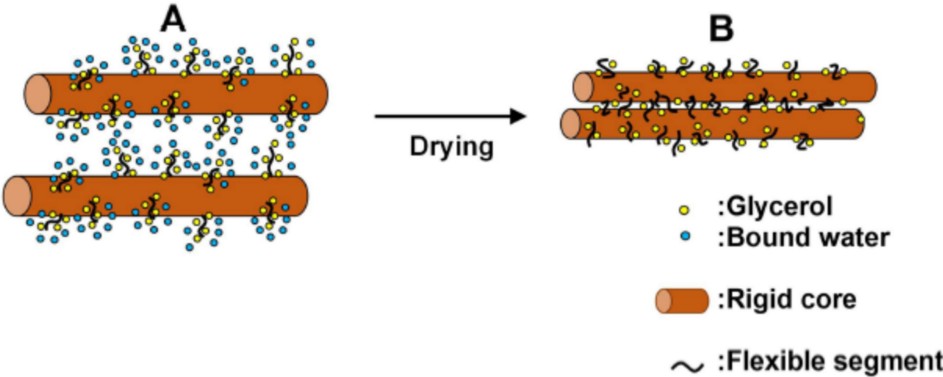

**Figure 13.** Structures, (**A**,**B**) of soft keratin, which is composed of a rigid core and flexible segments, under two conditions. Bound waters are attracted around flexible segments via natural moisturizing factors. When glycerol is applied, natural moisturizing factors are thought to be replaced by glycerol. (**A**). The case when glycerol was applied to the stratum corneum. (**B**). The case when stratum corneum treated with glycerol was dried.

In the X-ray diffraction experiment on the oriented SC sample, a broad X-ray diffraction peak for the 1-nm spacing of soft keratin was observed in the meridional plane [16], therefore by the X-ray diffraction we can detect the contribution of soft keratin in the rigid core in which the keratin fibers lie parallel to the long direction, where the 1-nm spacing of soft keratin is related to the distance between the two α-helix chains in the rigid core. However, the X-ray diffraction does not provide direct evidence of the non-helical flexible segments of soft keratin that play an important role in hydration within the corneocytes. As mentioned above, the structure of the non-helical flexible segments correlates with the α-helical structure in the rigid core, so that evidence in the non-helical region can be obtained indirectly from behavior in the α-helical region. The 1-nm spacing of soft keratin was measured as a function of the water content by the X-ray diffraction [36]. Increasing the water content from the dry SC, the spacing increases steeply and at about 25 wt% the change becomes gentle [36], i.e., the behavior of the 1-nm spacing is closely related with the change of the relative intensity of the nuclear magnetic resonance spectrum.

The structure of keratin in the SC contains α-helix and β-sheet. It has been found by the WAXD experiments that the distance between the two α-helix chains varies as a function of relative humidity (RH) [31], that is, the keratin interchain distance is insensitive to low RH, but sensitive to changes in hydration at RH >85% where the water content is 25 wt% at about RH 70%, while the β-sheet structure in the keratin filaments is unaffected by the variations in RH. On the other hand, Choe et al. [91] proposed from the results obtained from confocal Raman microscopy that water molecules intercalate between the sheets of β-sheet keratin, and β-sheet/α-helix keratin form ratio tends to decrease from the 100–40% SC depth but does not change significantly at 40–10% SC depth. Therefore, currently there is a contradiction in the influence of water molecules on the soft keratin structure in both reports. This point needs to be resolved in future research.

Outside the corneocytes, as mentioned in Section 2.4, the repeating distance of the SLS expands with the increase of water content, but that of the LLS does not. From small angle X-ray diffraction experiments, it was found that not only STS, but also LLS become stable with a water content of 25 wt%, that is, the water content of the SC is regulated to stabilize

both structures [11,12]. Currently, the relationship between the two lamellar structures is not clear.

As mentioned above, the water content 25 wt% plays an important role not only in water storage, but also in water content regulation, and even in barrier function. These molecular events seem to be independent. However, they may be entwined and work to maintain the skin normal on the common basis that the key water content is 25 wt%.

*4.2. Working Principle of Glycerol in Stratum Corneum as Moisturizer*

As mentioned in Section 4.1, there are at least two mechanisms for maintaining the normal water content of the SC. One is to store water in the SC. The other is to regulate the water content normally. In the former, soft keratin in the corneocytes plays an important role, and in the latter, the water layer of the SLS, which belongs to the same domain of the ORTHO [20], plays an important role [12,36]. Yamada et al. [32] investigated the role of aqueous solutions of polyols, glycerol and diglycerol, in the moisturizing effect by dynamic X-ray diffraction experiments for the first time at the molecular level. They focused on measuring the moisturizing effect of the polyols during the drying process in the SC. This is because one of the most serious conditions for the skin is exposure to dryness, and it is important to develop effective moisturizers to maintain normal skin even in dry conditions.

In the dynamic X-ray diffraction experiments, the SC samples were immersed in water or the polyol solutions at room temperature and the temporal variation in the spacings of soft keratin and the ORTHO became almost unchanged at 120 min. Dry $N_2$ gas was used to dry such treated samples and during the drying process the XRD measurement was performed using the solution cell [49]. In the drying process, 0.3 L min$^{-1}$ of dry nitrogen gas was flowed through the sample cell to dry the treated SC. The successive change of the diffraction peak of the 1-nm spacing of soft keratin and the spacings of the ORTHO were measured for 20 min. It was found that in the ORTHO, the spacings of the diffraction peak showed very small and complicated changes as follows. The overall changes of the of the spacings were only about 0.1%. Nevertheless, these changes during the drying process were universal and can be described in four distinct stages: in Stage I, the spacings decrease in a short time after starting to dry; in Stage II, they remain nearly flat; in Stage III, they increase in a short time; in Stage IV, they become almost flat. From the fact that the widths became narrow and the spacing became tight in Stage II, it was concluded that the Stage II is in a stable state. Including the stages partly being in the stable state, the total interval of Stage I, II and III was defined as the retention time of the stable state. The retention time was about 13 min for water and about 14 min for 10 wt% glycerol aqueous solution. There is no clear difference between them.

On the other hand, the 1-nm spacing of soft keratin decreased with time. Figure 14 shows typical results of the drying process after treating with (A) water and (B) 10 wt% glycerol aqueous solution. The descendent slope is 0.0011 nm/min for water and 0.0004 nm/min for 10 wt% glycerol aqueous solution. The glycerol aqueous solution strongly affects the structure of soft keratin and prevents rapid change.

In connection with the above results, it is well known that water storage in the SC is achieved by the presence of natural moisturizing factor (NMF) in the corneocytes [92,93]. The $^{13}C$ NMR study by Jokura et al. [28] provides clear evidence for the behavior of NMF in the corneocytes. As mentioned above, according to their argument, a keratin filament in the corneocytes is substantially divided into two regions, a non-helical amorphous region and an α-helical central core region. The NMF in the corneocytes serves two roles. Hydrated NMF has ionic interactions in the non-helical region of keratin fibers. Water molecules around the NMF form hydrogen bonds and this creates elasticity in soft keratin.

As mentioned above, the decay rate of the 1-nm spacing of soft keratin in the drying process are considerably different between the SC samples treated with water and 10 wt% glycerol aqueous solution. The evidence observed in the XRD experiment is due to changes in the distance between the two α-helix chains in the rigid core, but this structural change reflects the events that occurred in the flexible segment as schematically shown in Figure 13,

where the bound water in Figure 13A is maintained at 25 wt% in the normal SC. When drying the treated SC sample, in the beginning of the drying process the contribution of free water is probably undetectable by the XRD experiment since free water is removed quickly. Then, as the bound water is removed from soft keratin, it causes gradual decrease of the 1-nm spacing. Bound waters bind to the flexible segments of soft keratin via NMF, in the SC treated with water, bound water removed rather quickly. On the other hand, in the SC treated with 10 wt% glycerol aqueous solution, bound water removed slowly. From this fact, we can speculate that the replacement of NMF with glycerol suppressed the removal of the bound water, that is, glycerol interacts with flexible segment stronger than NMF. As a result, bound water is maintained for longer time in the glycerol treated SC. On the other hand, the retention time from the Stage I to III in the ORTHO is associated with the strength of the water regulation in the SC, since the ORTHO belongs to the same domain with the SLS [20], in which the water regulation was pointed out [12,36]. The retention time is longer in the SC treated with 10 wt% glycerol aqueous solution. Furthermore, this becomes further longer in the SC treated with 10 wt% diglycerol aqueous solution [32]. Therefore, the polyols work effectively to regulate the water content under the normal condition. In summary, the key point of the structural modification is that glycerol can replace NMF and achieve a normal skin condition, demonstrating greater capability in storing water and regulating the water content in the SC.

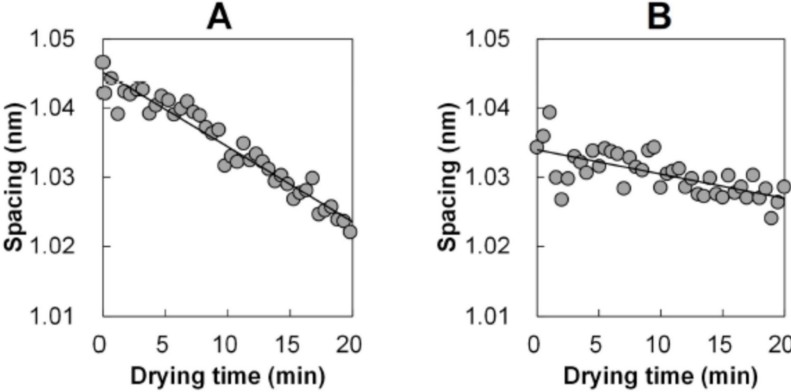

**Figure 14.** Change of 1-nm spacing of soft keratin as a function of time after drying the stratum corneum treated with (**A**) water and (**B**) aqueous solutions of 10 wt% glycerol. Adapted from ref. [32].

## 5. Liquid State in Intercellular Lipid Matrix Underlying the 500 Da Rule

Last but not least, on considering the penetration of drugs or cosmetics in the SC, it is important to take into account the liquid state (LIQUID), because the disordered structure should be one of the prominent penetration routes. White et al. [10] pointed out from the X-ray diffraction experiment in the murine SC that a broad X-ray diffraction peak at 0.45 nm is partly due to the LIQUID composed of the hydrocarbon chain, however, over the similar range a broad X-ray diffraction peak due to soft keratin lies, therefore it was difficult to the estimate of the proportion of the LIQUID in the intercellular lipid matrix. Doucet et al. [16] tried to estimate the proportion in such a way as to exclude the contribution of soft keratin in the broad peak at 0.45 nm. The human SC was partially delipidized by 1:1 chloroform-methanol mixture, and from the untreated broad peak the delipidized broad peak was subtracted. As a result, the remaining profile possesses only the contribution of the intercellular lipids composed of 0.41-nm sharp peak, the 0.37-nm sharp peak and the 0.46-nm broad peak. By employing the analysis of estimation of crystallized proportion [94], the proportion of the sharp peaks due to the ORTHO and the HEX was estimated to be 20% ± 10%, that is, the proportion of the LIQUID results in about 80%. To estimate the proportion of the LIQUID, it is necessary to extract the lipids in the SC uniformly. However, as pointed out in their paper [16], it might overestimate the proportion of the LIQUID, since the LIQUID might be extracted much more strongly in comparison

with the HEX and the ORTHO. On the other hand, from the result of the solid-state NMR for the porcine SC, it was proposed that most of the lipids are rigid [95], although the proportion of the ordered hydrocarbon-chain packing structures was not clarified. Therefore, this result is significantly inconsistent with the result obtained by X-ray diffraction [16]. We reconsidered the analysis by the X-ray diffraction from a different perspective.

It has been known that the intercellular lipids are extracted markedly by chloroform/methanol mixture [96–98], while ethanol extracts the lipids to a lesser extent [97–99]. Bommannan et al. [99] have proposed, based upon the results with ATR-FTIR, that ethanol extracts the intercellular lipids. However, ethanol does not cause a collapse of the ordered hydrocarbon-chain packing structures. Based upon these results we conducted a dynamic X-ray diffraction experiment when ethanol was applied to the human SC [17]. The original idea of this experiment is as follows. By applying ethanol to the SC for a sufficient time, the lipids of the LIQUID are mainly extracted. Therefore, we can obtain the X-ray diffraction profile without the contribution of the lipids of the LIQUID. Next, we subtract the profile of the SC sample after the ethanol treatment from the profile of the SC sample before the ethanol treatment. As a result, we can obtain the profile of the lipids of the LIQUID. From the profile of the LIQUID and the total profiles at the peaks of the 0.41 nm and 0.37 nm, we can finally obtain the ratio of the region of the LIQUID and the region due to the ORTHO and the HEX, i.e., we can obtain the ratio of the lipids of the LIQUID to the total lipids. In this process, to obtain the profile of LIQUID, it is necessary to remove the ethanol profile, which has a broad diffraction peak around S~2.5 nm−1 (0.4 nm). However, resultingly, we encountered unexpected phenomena.

As shown in Figure 15A,B, the integrated intensities of the sharp peaks at 0.41 nm and 0.37 nm did not change from the beginning to 11,000 s. This result is consistent with the result obtained from ATR-FTIR [99]. Therefore, ethanol is thought to have a significant effect on the LIQUID lipids, not the OTHO and the HEX lipids. However, as shown in Figure 15A, B, the integrated intensities of the 0.41-nm and the 0.37-nm sharp peaks increased significantly, in which the data are obtained at 6 and 22 h after removing ethanol are conveniently displayed on the 12,000 s position. However, the spacings of both peaks did not show change before the ethanol treatment, during the ethanol treatment and after the removement of ethanol. At 22 h almost all ethanol in the SC seems to be removed judging from the increasing behavior of the integrated intensities. This is consistent with the result obtained by van der Merwe and Riviere [100] that ethanol is not present in the SC treated with ethanol after drying 24 h. The above fact indicates that, although ethanol does not affect the ORTHO and the HEX, ethanol dissolves the lipids of the LIQUID and then, by removing ethanol from the SC the ORTHO and the HEX are emerged by crystallization from the mixture of the ethanol and the LIQUID. Sum up the above event in the illustration of Figure 16A,B. In Figure 16A, when the SC is exposed in ethanol, the ORTHO and the HEX are not affected, but the lipids in the LIQUID are dissolved in ethanol, therefore ethanol and the LIQUID results in mixture. In Figure 16B, after removing ethanol from the ethanol-treated SC, the ORTHO and the HEX emerge newly from the mixture. Therefore, the proportion of the LIQUID is reduced significantly.

Based upon the above result of the crystallization, we tried to estimate the proportion of the LIQUID [17]. This gives only the minimum value of the proportion of the LIQUID, because a part of the LIQUID produces the ordered hydrocarbon-chain packing structures and some part of the LIQUID is extracted by ethanol. To estimate the minimum proportion of the LIQUID, the integrated intensities of the 0.41 nm peak and the 0.37 nm peak were analyzed: Before exposing ethanol the integrated intensities of 0.41 nm peak and 0.37 nm peak are given by $I^{0.41}_{\text{pre-treated}}$ and $I^{0.37}_{\text{pre-treated}}$, respectively; after removing ethanol for 22 h from the ethanol-treated SC they are given by $I^{0.41}_{\text{post-treated}}$ and $I^{0.37}_{\text{post-treated}}$, respectively. The ratios, $I^{0.41}_{\text{post-treated}}/I^{0.41}_{\text{pre-treated}}$ and $I^{0.37}_{\text{post-treated}}/I^{0.37}_{\text{untreated}}$, are calculated. From the measurement on three donors, $I^{0.41}_{\text{post-treated}}/I^{0.41}_{\text{pre-treated}}$ was calculated to be 1.13 to 1.88 and $I^{0.37}_{\text{post-treated}}/I^{0.37}_{\text{untreated}}$ was calculated to be 1.28 to 1.93. Since it is reasonable to adopt the maximum value for the estimation of the proportion of the LIQUID, the largest

value of $I^{0.41}_{\text{post-treated}}/I^{0.41}_{\text{pre-treated}}$ and $I^{0.37}_{\text{post-treated}}/I^{0.37}_{\text{untreated}}$ is close to 2. Therefore, the proportion of the lipids of the LIQUID is estimated to be greater than 50%. As aforementioned, from the results proposed by Doucet et al. [16] the proportion of the lipids of the LIQUID is less than 80% of the total intercellular lipids. Therefore, we propose that the proportion of the lipids in the LIQUID lies between 50% and 80% in the whole intercellular lipids. Therefore, the region occupied by a large amount of the disordered LIQUID should play an important role in the penetration of substances through the SC.

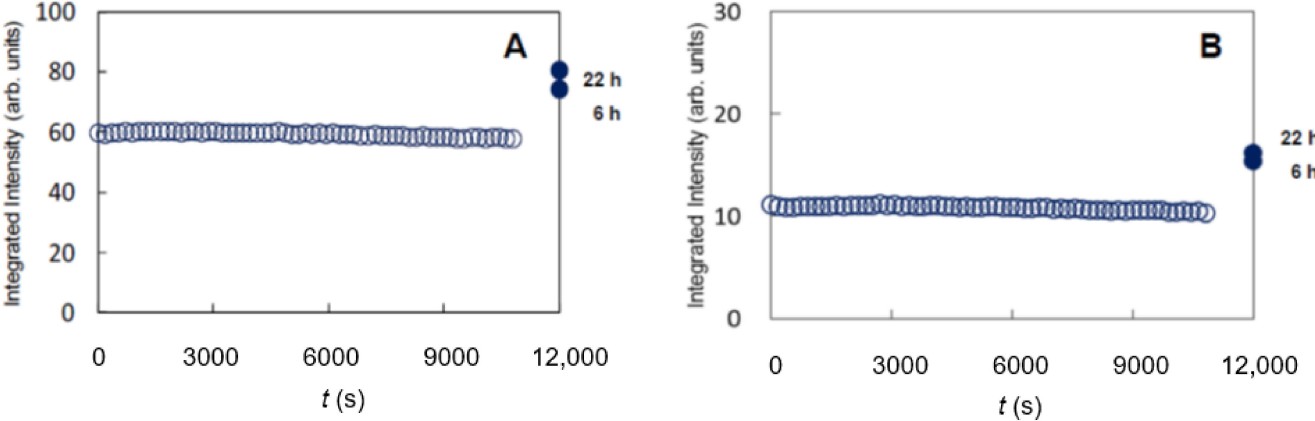

**Figure 15.** Change of integrated intensities at 0.41-nm and 0.37-nm peak in the hydrocarbon-chain packing structure of the stratum corneum treated with ethanol. (**A**). Change of the integrated intensity at the lattice constant 0.41 nm. (**B**). Change of the integrated intensity at the lattice constant 0.37 nm. During applying ethanol to the stratum corneum the 0.41-nm and 0.37-nm peaks are almost unchanged until 10,000 s. For convenience, the data denoted on the time scale at 12,000 s were obtained by removing ethanol from the ethanol-treated stratum corneum for 6 h and 22 h. By removing ethanol in the mixture of the intercellular lipids and ethanol, the integrated intensities increase due to crystallization of the ordered hydrocarbon-chain packing structures.

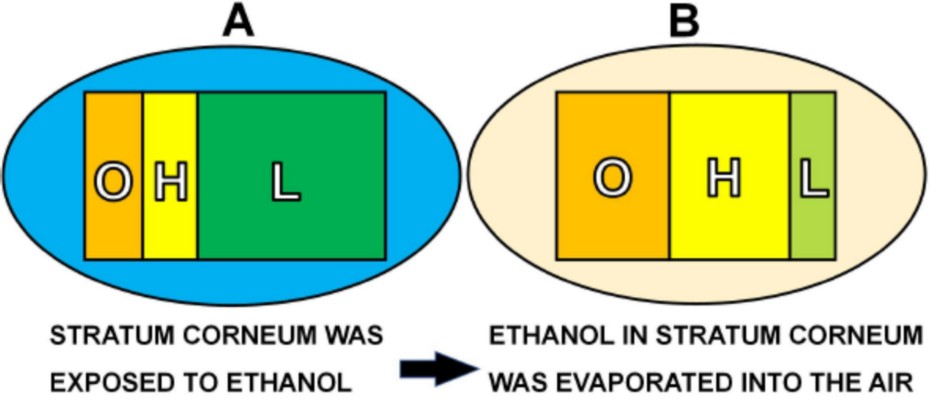

STRATUM CORNEUM WAS EXPOSED TO ETHANOL → ETHANOL IN STRATUM CORNEUM WAS EVAPORATED INTO THE AIR

**Figure 16.** Schematic of the ratio of regions with the orthorhombic (O), the hexagonal (H) hydrocarbon-chain packing structures and the liquid state (L) in the intercellular lipid matrix. (**A**). The ratio during immersing in ethanol. Ethanol is shown by blue area. The disordered hydrocarbon-chain packing structure dissolved in ethanol is shown by dark green area, where the ordered hydrocarbon-chain packing structures are not affected by ethanol application. (**B**). The ratio after removing ethanol from the stratum corneum. The areas denoted by O and H increase. The area denoted by L shown by light green decreases.

We further consider the role of the LIQUID that should play an important role in percutaneous penetration. Until now, for the penetration of hydrophobic molecules, the 500 Da rule has been proposed by Bos and Meldardi [101], who argued that the molecular weight (MW) must be less than 500 to allow the penetration in skin and that larger molecules

cannot penetrate. In opposition to this proposal, Roberts et al. [102] argued that, when they examined molecules over 500 Da, this was not the case. More precisely, Potts and Guy [103] analyzed the data set (more than 90 solute compounds with MW ranging from 18 to over 790, and $\log K_{o/w}$ ranging from $-3$ to 60) and proposed that the permeabilities are expressed by an elegant empirical relation of MW vs. $\log K_{o/w}$ that also applies to molecules above 500 Da. Therefore, it is interesting to know the relationship between the 500 Da rule and the existence of a large region composed of the LIQUID.

The average MW of lipids in the LIQUID is important in studies on the 500 Da rule. It is inferred that the average molecular weight of the most predominant molecules in the LIQUID provides a measure of the MW of the foreign molecule that can easily penetrate the SC. In the human stratum corneum, the intercellular lipids are dominantly composed of CERs (50 wt%), CHOL (25 wt%), and FFAs (15 wt%) [104,105]. Among them, the MW of CHOL is 387 Da. In the human SC, the relative abundance of FFAs is highest at the chain length of carbon number 24 [106,107]. Therefore, the central MW of FFAs is estimated to be about 369 Da ($C_{24}H_{48}O_2$). The relative abundance of CERs is highest at the chain length of carbon number 46 and that of CER subclasses is highest at CER NP [24]. Therefore, the central MW of CERs is estimated to be about 724 Da ($C_{46}H_{93}NO_4$). From the sum of the central MWs weighted by their population, we could calculate the average MW of the intercellular lipids in the LIQUID. From this calculation, the average MW became about 514. Based upon this result, when a hydrophobic molecule of about 500 Da is applied to the surface of the SC, it is proposed that it merges to the lipids in the LIQUID, then permeates and/or diffuses into the intercellular lipid region with the LIQUID. The much smaller lipophilic molecule diffuses much easier, but on the other hand the bigger molecule becomes difficult to diffuse since the diffusion constant becomes smaller. In addition, since the intercellular lipid molecules have alkyl chains, it can be inferred that the LIQUID is suitable for the permeation of foreign molecules having alkyl chains. The functional groups of the intercellular lipids may also aggregate around foreign molecules to form locally stable conformations. Therefore, the maximum molecular size that can penetrate the SC is not necessarily 500 Da, but will depend on physicochemical properties, such as $\log K_{o/w}$, molecular interaction, etc. Finally, I stress that, since the ordered structures, the HEX and the ORTHO, and the LLS and the SLS, have their own different functions, a characteristic molecule could possibly penetrate through the specific route in the ordered structures.

## 6. Summary

The X-ray diffraction study used to solve the working mechanism of various functions in the stratum corneum was discussed in typical subjects as follows: Damage to the long-period lamellar structure was caused by sodium dodecyl sulfate, and such damaged stratum corneum was reconstituted by applying acylceramide. Applying glycerol to the dry stratum corneum maintained normal conditions for a long period of time. From the analysis of structural modification of the intercellular lipid structures when applying ethanol to the stratum corneum, the existence of a large proportion with disordered intercellular lipid structure was proposed. Based on this result, the mechanism of the so-called 500 Da rule in drug or cosmetic penetration was discussed. In addition, for the other subjects, including the key water content 25 wt%, penetration of drug-loaded nanoparticles, etc., please see another review article recently written [18].

**Funding:** This work was supported by a Grant-in-Aid for Scientific Research (C) (15540397) and (C) (18540411) from the Ministry of Education, Culture, Sports, Science and Technology, Japan.

**Institutional Review Board Statement:** This study was conducted in accordance with the Declaration of Helsinki and approved by the Ethics Committee of Nagoya Industrial Science Research Institute.

**Informed Consent Statement:** Human stratum corneum samples obtained from Biopredic International were used. This company collected samples from patients who signed an informed consent form.

**Data Availability Statement:** The data presented in this study are openly available in Figures 1–3 at doi:10.5650/jos.ess21159, ref. [19]; in Figures 2–4 at Chemistry and Physics of Lipids **123** (2003) 1-8, ref. [12]; in Figures 1 and 3 at doi:10.1111/ics.12430, ref. [69]; in Figure 2 at SPring-8 Research Frontiers 2017 (2018) 90–91, in ref. [70]; in Figures 6 and 7 at doi:10.1016/j.chemphyslip.2018.05.003. ref. [90]; in Figure 3 at doi:10.1111/ics.12664, ref. [34]; in Figures 2 and 3 at doi:10.3390/pharmaceutics9030026, ref. [18].

**Acknowledgments:** Special thanks to J. A. Bouwstra (Leiden University), S. Ban (Nippon Menard Cosmetic), A. Habuka (Sakamoto Yakuhin Kogyo), G. Imokawa (Utsunomiya University), A. Nakaune-Iijima (FUJIFILM), H. Nakazawa (Kwansei Gakuin University), N. Ohta (SPring8/JASRI), A. Sugishima (FUJIFILM), H. Tanaka (Nippon Menard Cosmetic), T. Tashiro (FUJIFILM), N. Yagi (SPring8/JASRI), T. Yamada (Sakamoto Yakuhin Kogyo), K. Yanase (Kracie Home Products) for their fruitful discussions and/or earnest cooperation and other collaborators who are interested in subjects similar to this paper. The dynamic X-ray diffraction experiment was performed on SPring-8. We would like to thank JASRI for using the SPring-8 facility.

**Conflicts of Interest:** The author declares no conflict of interest.

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
