# Peer review of "Stratum Corneum Structure and Function Studied by X-ray Diffraction"

_dermato, doi:10.3390/dermato2030009_

Round 1
Reviewer 1 Report
The author reviews the X-ray diffraction experiment in stratum corneum structure and the function studied. I am surprised that the author did not mention the application of X-ray diffraction in a topical drug delivery study. Also, the TEWL meter and ATR-FTIR are famous for barrier function study. The author may compare the advantages and disadvantages of those tools.
Author Response
Comments of Reviewer 1
The author reviews the X-ray diffraction experiment in stratum corneum structure and the function studied. I am surprised that the author did not mention the application of X-ray diffraction in a topical drug delivery study. Also, the TEWL meter and ATR-FTIR are famous for barrier function study. The author may compare the advantages and disadvantages of those tools.
Reply from author
Thank you for your valuable comments. Especially I understood that I have to pay attention to readers in the broad research area of skin as a review paper.
- In this manuscript, a topical drug delivery study was not directly mentioned. The related subject was written in Chapter 3 “Liquid State in Intercellular Lipid Matrix”. However, it is not clear whether this title is concerned with drug delivery. I modified the title of Chapter 3 to indicate clearly this point:
-Line 937 in page 23: The title is revised as “Liquid State in Intercellular Lipid Matrix Underlying the 500 Da Rule”.
-Line 1084-1086 in page 26: And I recently wrote a review paper in J. Oleo Sci., in which penetration of drug-loaded nanoparticles were discussed. I added this fact.
- In study on barrier function, I agree that TEWL and ATR-FTIR certainly are indispensable methods. Comment a) of Reviewer 2 is also related to this point. In the manuscript, these methods were described as follows:
-Line 239-249 in page 7; Line 563 in page 14: TEWL is one of the complementary methods in the study of barrier function. To clarify this point, the usefulness of TEWL and electrical conductivity and capacitance measurements was discussed on a case-by-case basis as the experimental results were described.
-Line 86-91 in page 2; Line 946 in page 24; Line 963 in page 24; the usefulness of ATR-FTIR was discussed also on a case-by-case basis as the experimental results were described.
-Line 42-58 in page 1-2: To study the structure in similar scat to X-ray diffraction, there are several methods. They have the advantages and disadvantages. Comparison of the advantages and disadvantages was made among X-ray diffraction, neutron diffraction and electron diffraction. There are a lot of observation methods on the structures from small scale to large scale. The interpretation on electron microscopy [8], optical microscopy and fluorescent microscope was also added. In addition, confocal Raman microscopy is a very useful method as discussed in the following interpretation.
-Line 36 in page 1; Line 745 in page 19: Confocal Raman microscopy is a unique tool for obtaining the water content in in vivo SC. Based on the results obtained by this method, the behavior of water within the SC can be further discussed.
Reviewer 2 Report
Ichiro Hatta. Stratum Corneum Structure and Function Studied by X-Ray Diffraction
For several years the topic about the skin barrier and the stratum corneum structure and its function consistently is developed. As it is presented in the current form, the manuscript is interesting, essential, well organized and clear. In the reviewer's opinion, before to be accepted for publication on Dermato Journal (MDPI), the author should consider a couple of minor revision points:
Introduction:
The cited information and data are already well described in the scientific literature, so it would be worth:
a) Specify what additional information is possible to obtain only by the X-ray diffraction method, which cannot be obtained by other technique;
b) Provide a table in which the details about the stratum corneum structures and sizes of them will be presented which were received by various techniques. It will allow to compare the achievements of knowledge by using different methods;
c) Give the distances between the ceramide chains in the lipid matrix and what spaces in [nm] can be formed when cholesterol and free fatty acid molecule are presented among ceramides;
d) Describe, what is usually the percentage of beta-sheet in relation to alpha-helix in corneocytes?
How does this ratio change as a result of interaction with a water or active substance molecule?
Which conformation of keratin is more stable? Is it possible to prove by some methods?
What conformation: beta-sheet or alpha-helix impact on more skin barrier?
Author Response
Comments of Reviewer 2
For several years the topic about the skin barrier and the stratum corneum structure and its function consistently is developed. As it is presented in the current form, the manuscript is interesting, essential, well organized and clear. In the reviewer's opinion, before to be accepted for publication on Dermato Journal (MDPI), the author should consider a couple of minor revision points:
Introduction: The cited information and data are already well described in the scientific literature, so it would be worth:
a) Specify what additional information is possible to obtain only by the X-ray diffraction method, which cannot be obtained by other technique;
b) Provide a table in which the details about the stratum corneum structures and sizes of them will be presented which were received by various techniques. It will allow to compare the achievements of knowledge by using different methods;
c) Give the distances between the ceramide chains in the lipid matrix and what spaces in [nm] can be formed when cholesterol and free fatty acid molecule are presented among ceramides;
d) Describe, what is usually the percentage of beta-sheet in relation to alpha-helix in corneocytes?
How does this ratio change as a result of interaction with a water or active substance molecule?
Which conformation of keratin is more stable? Is it possible to prove by some methods?
What conformation: beta-sheet or alpha-helix impact on more skin barrier?
Reply from author
Thank you for your valuable comments.
- For the information obtained by other techniques, as mentioned in Reply 2 to Reviewer 1, these were shown in the manuscript on a case-by-case basis as the experimental results were described. I hope that it is one of the ways to know deeply the usefulness of these techniques.
- These points were discussed in the manuscript on a case-by-case basis as connected with above reply a).
- Line 102-142 in page 3-4: Lateral distances between the lipid molecules were calculated based on the lattice constants. The behavior of cholesterol in the formation of the lateral packing structures remains to be clarified. To obtain the basic structural evidence, the molecular size of the long axis of the hydrocarbon chain was calculated including cholesterol. From this consideration, it seems that the cholesterol presence and its shape in the intercellular lipid matrix seems to be a little clearer.
- Line 845-855 in page 21: Discussion for α-helix and β-sheet was introduced.
Round 2
Reviewer 1 Report
The author has made sufficient revisions. The reviewer would suggest that the author should add a table to show/summarize the suitable application situation, advantages/disadvantages by using those different instruments such as X-ray, ATR-FTIR, Raman microscope, and TEWL in dermatology research.
Author Response
Reply to Reviewer 1
Thank you for your suggestions.
According to the suggestions, I added Table 1 from Line 41 to 48 in the revised manuscript.
Thank you again. I could clearly describe the advantages and disadvantages of the different methods.